# Review of Machine-Learning Techniques Applied to Structural Health Monitoring Systems for Building and Bridge Structures

**Alain Gomez-Cabrera and Ponciano Jorge Escamilla-Ambrosio ***

Centro de Investigación en Computación, Instituto Politécnico Nacional, Mexico City 07738, Mexico
* Correspondence: pescamilla@cic.ipn.mx

**Abstract:** This review identifies current machine-learning algorithms implemented in building structural health monitoring systems and their success in determining the level of damage in a hierarchical classification. The integration of physical models, feature extraction techniques, uncertainty management, parameter estimation, and finite element model analysis are used to implement data-driven model detection systems for SHM system design. A total of 68 articles using ANN, CNN and SVM, in combination with preprocessing techniques, were analyzed corresponding to the period 2011–2022. The application of these techniques in structural condition monitoring improves the reliability and performance of these systems.

**Keywords:** structural health monitoring; machine learning; physics-based model; data-based model; building structures





## 1. Introduction

Structural health monitoring (SHM) is a field of science that focuses its efforts on evaluating and monitoring the integrity of a structure of interest [1]. Structural health monitoring systems are based on the design of sensing systems and structural models to evaluate machines and structures.

Although SHM systems are not a new field of research, computational advances in sensing hardware and the computational power of embedded devices drive the generation of reliable data for developing models based on classification and prediction data, including machine-learning algorithms in SHM systems. Moreover, sensors, such as accelerometers, are inexpensive compared to other sensors and can be effectively deployed in a sensing system to implement vibration-based SHM systems [2]. Accelerometers, or when combining them with other sensors, are the dominant sensing approaches for these SHM applications [3]. Because vibration-based systems date back to the late 1970s [4], technological advances represent a field of opportunities to improve existing solutions in the field of damage identification.

Previous studies have been conducted in the field of SHM using machine-learning (ML) techniques. However, the importance of this study is to compare the application of artificial neural networks (ANNs), convolutional neural networks (CNNs) and support vector machine (SVM) techniques considering the input data, feature selection techniques, the structure of interest, data size, the level of damage identification and the accuracy of the ML model. In addition to the fact that each of the above ML techniques has been used for similar tasks in structural damage and system identification, some of them perform better when the data comes from data generated by multi-sensor data fusion or when the data are processed with damage-sensitive feature extraction techniques, such as Hilbert–Huang transform (HHT) or wavelet packet transform (WPT). In addition, this study could provide a starting point for the selection of ML techniques and signal processing techniques for future SHM ML-based solutions where structural configuration or data features have similarity with previous studies that achieved good structural damag or system identification performance.

This paper is structured as follows. First, a brief description of the concepts related to SHM systems is presented in Section 2. In Section 3, the most common feature selection algorithms are presented, and a review of related works is given. In Section 4, the two main branches of SHM system models are presented, namely, the physics-related model (Section 4.1) and the data-driven models (Section 4.2). In Section 4.2, three ML techniques and their application in SHM systems are explored: ANN, CNN and SVM. At the end of Section 4, a comparison between the two main model branches is established in Section 4.3. Section 5 includes a brief review of the use of signal processing techniques to mitigate uncertainties effects from the environment and noise. In Section 6, a summary of the most important features of the application of damage identification techniques are presented in a list. Finally, Section 7 contains conclusions and future work.

## 2. Damage Classification in SHM

SHM covers several application areas and the assets monitored range from small components to huge civil structures and complex machines. Building SHM systems focus on measuring changes in the physical parameters to assess the current state of the structure and, in some cases, predict the building's response to future seismic excitations. To make these predictions, it is necessary to identify the natural frequencies of the buildings [5]. In the case of buildings, the structure is subjected to the effects of static and dynamic loads, so the complexity of the analysis presents a challenge in giving an accurate model that includes all these known and unknown effects.

SHM systems are composed of several hardware and software elements. An overview of the main components of SHM systems as defined by Farrar and Worden [6] are:

1. Operational assessment: The aspect related to damage conceptualization and operational conditions.
2. Data acquisition: The sensor system design and data preprocessing.
3. Feature extraction: The selection of sensitive damage features according to the damage identification capabilities of the desired SHM system.
4. Statistical model development: The design and implementation of the physics-based or data-based model.

Yuan et al. [1] explored the data acquisition aspect (1) of recent proposals for SHM systems, showing several features of accelerometer sensing systems that are attractive for the structural monitoring and evaluation of SHM systems. This study focuses on areas three and four of SHM systems, analyzing proposals of the physics-based models and the data-based models presented in the literature that belong to these areas. Figure 1 shows these areas in SHM systems, and the methods, techniques and algorithms involved.

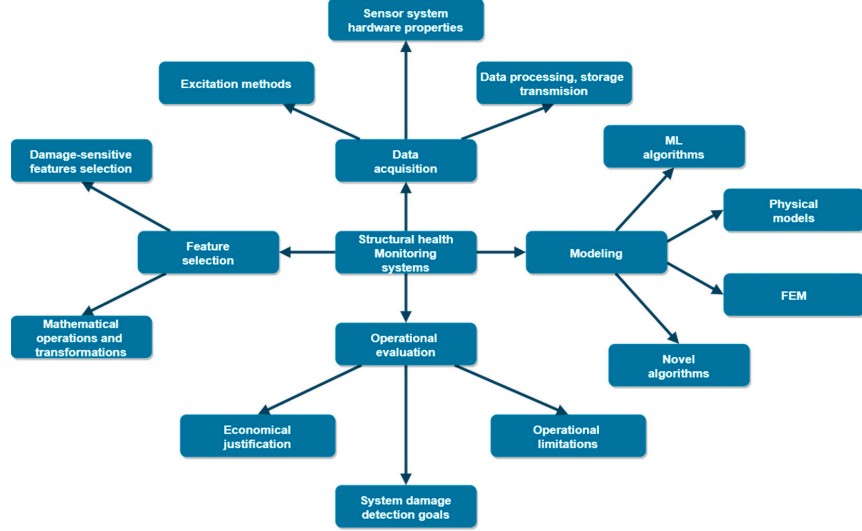

**Figure 1.** Diagram of structural health monitoring systems areas.

The core of SHM systems is their ability to perform damage identification. Damage is the change in the material's physical properties due to progressive deterioration or as a result of a single event on a structure. This change can detract from the behavior or integrity of a structure. Damage characterization can be conceived in several ways depending on the objectives of the SHM system, and damage states can be defined in terms of extent, severity, remaining operational time, thresholds, and damage index standards. Rytter [7] also presents a damage identification classification in SHM systems, as shown in Table 1.

**Table 1.** Damage characterization levels.

| Damage Characterization Level | Description |
| --- | --- |
| I: Detection | The SHM system can decide if there is any damage to the structure of interest. |
| II: Localization | The SHM system can determine the existence and location of damage in the structure of interest. |
| III: Assessment | The SHM system can estimate the extent of damage in the structure of interest. |
| IV: Prediction | The SHM system can estimate the remaining lifetime of the structure. |

## 3. Feature Selection Algorithms

In the field of SHM, several signal processing techniques can be used to extract damage-sensitive features and detect, locate and estimate structural damage in the machines and structures of interest. Some of the techniques used in previous works in this field include ANFIS [8,9], wavelet-based techniques [10], frequency–response functions (FRF) [11], Kalman filter-based techniques [12] and soft computing-based techniques [13].

The selection of damage-sensitive features by manual means represents a problem, as it requires expertise to identify candidate damage-sensitive signals. Algorithms for producing these features have been explored in the literature and two approaches to the generation of these features are widely proposed. One common approach is to propose novel features based on signal processing techniques. Another is to let a machine model the task of automatically extracting some sensitive features from data hidden on the surface. Composite damage indicators can also be constructed from the combination of damage-sensitive features. For example, Lubrano Lobicanco et al. [14] used residual drift ratios and lateral stiffness to define damage levels and quantify the amount of damage to structures due to seismic events.

A comprehensive review of some damage-sensitive feature construction algorithms can be found in the review of Civera and Surace [15]. Among the most recent techniques applied to the problem of extracting useful information from data for structural damage identification are principal component analysis (PCA), the wavelet transform (WT), along with its variants, and the application of empirical modal decomposition (EMD) in combination with the Hilbert–Huang transform (HHT).

### 3.1. Principal Component Analysis

Principal component analysis (PCA) is a statistical method mainly applied to reduce data dimensionality, which keeps relevant information from the original data. PCA application in ML models improve training time and model performance because redundant information in training data is removed. Initial data are projected into new variables, named principal components, with the most data variance. PCA use in SHM systems reduces large sensor-gathered data processing times, thereby enhancing the system response for structural changes or events resulting from structural damage.

Table 2 shows a compilation of recent works incorporating the PCA technique on the data for subsequent use by a machine-learning model. As can be seen, PCA is a popular technique in recent years due to the advantages that it offers. Although PCA is a powerful

tool for ML applications, the PCA computation calculates the eigenvalues and eigenvectors of the data covariance matrix, which is an expensive operation for large data dimensions.

### 3.2. Wavelet Transform

The wavelet transform (WT) decomposes a signal into a series expansion where the number of wavelet functions is multiplied by a set of coefficients representing the original signal. Similar to Fourier transform (FT), WT expresses a signal in more specific located-in-time functions, but in the case of WT, the original signal can be non-periodic. In particular, WT analysis of a signal can bring frequency information and its position in time. Wavelet-based techniques include the discrete wavelet transform (DWT), the wavelet packet transform (WPT), the continuous wavelet transform (CWT), and the stationary wavelet transform (SWT). Table 3 shows the combination of WT-based techniques with ML models for structural damage identification. In data-based models, WT techniques serve as dimensionality reduction techniques, noise removal filters, and damage-sensitive feature generators, thereby improving SHM systems' accuracy and robustness.

### 3.3. Empirical Mode Decomposition

Empirical mode decomposition (EMD) is one of the most used signal decomposition techniques in SHM. Several proposals exist in structural damage identification, modal parameters identification and structural prognosis. The advantages of using EMD over nonlinear and non-stationary signals, such as structural responses due to mechanical excitation, are the time–frequency information of the analyzed signal in all the signal lengths. Another advantage of the application of EMD in comparison to other signal decomposition techniques is that EMD is signal-adaptive.

Over time, variations have been proposed to tackle the limitations and issues of EMD applications, such as noise and the well-known mode-mixing problem. Ensemble EMD (EEMD), multivariate EMD (MEMD) and time-varying filter-based EMD (TVF-EMD) are used in combination with ML models to achieve better accuracy in structural damage and system identification, as can be seen in Table 4.

A systematic comparative study between the above-mentioned EMD variants can be found in [16], proving EMD to be a robust signal processing technique in structural damage identification, system identification and anomaly detection. Regardless of the benefits of using EMD variations, including mode-mixing minimization and noise removal, the computation of these algorithms can be prohibitively expensive in SHM real-time applications.

**Table 2.** PCA review.

| Publication | Structure | Excitation Source | Signal Processing and/or Feature Extraction Technique | Dataset Size | Purpose | Year |
|---|---|---|---|---|---|---|
| [17] | Instrumented bridge | Operational conditions | FFT | 43,200,000 data points | Improve anomaly detection accuracy and reduce the size of SHM transmitted data | 2021 |
| [18] | Steel-supported experimental beam model | Impact hammer | FRF | 16,384 data points | Study the relation between PCA-generated data to modal analysis and frequencies to establish a physical interpretation of PCA-compressed data | 2020 |
| [19] | The Phase II IASC-ASCE SHM benchmark structure FE model | Simulation | MSD | 480,000 data points | Propose a new method to extract damage-sensitive features that are invariant to environmental effects. | 2020 |

**Table 2.** *Cont.*

| Publication | Structure | Excitation Source | Signal Processing and/or Feature Extraction Technique | Dataset Size | Purpose | Year |
|---|---|---|---|---|---|---|
| [20] | Simply supported steel truss bridge experimental model | Impact hammer | FRF, PCA | 1200 samples of 512 data points | Propose a combination of PCA and FRF to obtain a new damage index for structural damage classification | 2020 |
| [21] | Benchmark concrete beam numerical model | Simulation | ARM, LDA, QDA, NB, DT | 8002 data points | Present a comparative study between PCA and AR models to extract damage-sensitive features and several classification methods | 2019 |
| [22] | Truss bridge FE model and two-story frame FE models | Simulation | FRF, LUT | Not specified | Propose a combination of PCA and LUT to train an ML model for structural damage detection and localization | 2017 |
| [23] | Truss bridge and two-story frame FE models | Simulation | FRF, 2D-PCA, ICA | 49 samples (bridge) and 75 samples (two-story frame) | Propose a structural damage detection, localization and severity estimation using a PCA-extracted damage index and an ANN | 2017 |
| [24] | Twin-tower steel structure experimental model | Shaking table | FRF | Not specified | Propose a scalogram-based damage detection and localization system using PCA and FRF | 2017 |
| [25] | Steel plate experimental model | PZT burst signal | Golay filter algorithm | 8 samples of 10,000 data points | Study the noise filtering capabilities of PCA and Golay algorithms to train an ML model for structural damage identification | 2016 |
| [26] | Thin aluminum plate experimental model | PZT burst signal | ARM | 1000 samples | Propose methodology to detect structural changes based on statistical inference | 2016 |
| [27] | Three-story frame aluminum structure experimental model | Bumper | AANN, FA, MSD, SVD, SVDD, KPCA | 17 samples of 4096 data points | Propose a comparative study between kernel algorithms for structural damage detection purposes | 2016 |
| [28] | Three-story frame aluminum structure experimental model | Shaking table | ARM | 1700 samples | Compare the performance of four PCA-based algorithms for structural damage detection under environmental conditions | 2015 |
| [29] | Grid FE model and steel grid structure experimental model | Simulation | PCA, WGN | 4000 samples | Propose an ANN SHM system for damage detection due to temperature changes and noise perturbations | 2015 |
| [30] | Small aluminum plate experimental model | PZT burst signal | Chi-square goodness-of-fit test | 600 samples | Propose a statistical analysis for structural damage detection using PCA-compressed data | 2014 |

**Table 2.** *Cont.*

| Publication | Structure | Excitation Source | Signal Processing and/or Feature Extraction Technique | Dataset Size | Purpose | Year |
|---|---|---|---|---|---|---|
| [31] | 12-DOF mass–spring-damped experimental model | Impact hammer | PCP | 20,000 samples | Propose a denoising method using PCP for SHM systems | 2014 |
| [32] | Building structure experimental model and Sydney Harbour-instrumented bridge | Electrodynamic shaker and operational conditions | RP, FT, SVM, PAA | 3240 samples of 8192 data points (building), 6370 samples of 3600 data points (bridge) | Propose a comparative study between kernel algorithms for structural damage detection purposes | 2014 |
| [33] | Aluminum plate experimental model | Piezoelectric transducers attached to the plate | SOM | 750 samples | Propose an SHM model for structural damage detection and classification | 2013 |
| [34] | Instrumented bridge | Numerical simulation | HHT, EMD | 500 samples | Propose an environmental noise removal method to improve ML model damage detection capabilities | 2013 |
| [35] | 10-story low-rise building FE model | Simulation | FRF, PCA | 240 samples | Propose a structural damage detection method using PCA for data dimensionality reduction and noise removal | 2013 |
| [36] | Aluminum plate experimental model | PZT burst signal | CDLM | 300 samples | Present a comparative study between five different methods for structural damage detection and localization | 2011 |
| [37] | Aluminum plate experimental model | PZT burst signal | Chi-square distribution | 100 samples | Propose a new structural damage detection methodology based on the probability distribution of PCA projections | 2011 |
| [38] | Supported steel beam experimental models | Impact hammer | FRF | 8192 data points | Propose a structural damage identification method using residual FRF data in combination with ANN | 2011 |

**Table 3.** WT review.

| Publication | Structure | Machine-Learning Algorithm | Dataset Size | Accuracy | Purpose | Year |
|---|---|---|---|---|---|---|
| [39] | Three-story building structure experimental model and steel frame experimental model | DCNN | 2280 samples (building) and 2000 samples (steel frame) | ACC = 99% (building), ACC = 99.610% (steel frame) | Generate 2D training samples using CWT from accelerations scalograms | 2021 |
| [40] | Aluminum plate FE model and aluminum plate experimental model | CNN | 300,000 data points | ACC = 99.9% | Wavelet transform was applied to acceleration signals to obtain wavelet coefficient matrix (WCM) in order to train a CNN | 2019 |

**Table 3.** *Cont.*

| Publication | Structure | Machine-Learning Algorithm | Dataset Size | Accuracy | Purpose | Year |
|---|---|---|---|---|---|---|
| [41] | Offshore platform FE model and scaled experimental model | TF, WPD, PCA | 120 samples of 10,000 data points (FE model), 48 samples of 17,500 data points (scaled model) | ACC = 78.125% (FE model), ACC = 84.375% (instrumented model) | WPE and low-order principal components were applied to structural response for SVM model training | 2018 |
| [42] | IASC–ASCE SHM benchmark four-story steel structure FE model | WPD | 99 samples | ACC = 91.75% | WPE was used to extract damage-sensitive features from accelerations signal to train a damage detection SVM model | 2015 |
| [43] | IASC–ASCE SHM benchmark four-story steel structure FE model | LS-SVM, PSO, HSA, SF, WPT | 16 samples of 40,000 data points | ACC = 100% (PSHS), ACC = 98.2% (Harmony), ACC = 98.68% (PSO) | Damage features were extracted using WPE in order to train an ANN and an LS-SVM model | 2014 |
| [44] | IASC–ASCE SHM benchmark four-story steel structure FE model | ANN | 621 samples | ACC = 95.49% (damage case 1), ACC = 96.78% (Damage case 2) | WPT was used to remove noise and interferences from structural responses signal to build a training dataset for an ANN damage detection model | 2012 |
| [45] | Four-story building FE model | BP-NN, WPT, Battle–Lemarie decomposition, WGN | 183 samples | ACC = 90% (single sensor), ACC = 100% (multiple sensor) | WPRE was used to extract damage-sensitive features from acceleration data in order to train an ANN damage detection model | 2011 |

**Table 4.** EMD review.

| Publication | Structure | Machine-Learning Algorithm | Accuracy | Purpose | Year |
|---|---|---|---|---|---|
| [46] | Mass–spring system numerical model and instrumented bridge | MSD, RARMX | Error graph | EMD method and IMF were used to build damage feature vectors | 2020 |
| [47] | Fourteen-bay steel truss bridge experimental model | ANN, CEEMD, HHT, GWN | RMSE = $5.156 \times 10^{-4}$ | A combination of CEEMDAN and HHT was proposed to extract four key damage-sensitive features to train an ANN model for damage detection, localization and severity estimation | 2020 |
| [48] | Simply supported steel Warren truss instrumented bridge | Neuro-fuzzy ANN | ACC = 80% | EMD method was used to reduce data dimensionality of SHM system for structural damage assessment | 2020 |
| [48] | Hanxi bridge FE model | FastICA | Correlation = 0.84 | EEMD and PCA were used to extract independent deflection components from structural responses and determine the presence of damage | 2018 |
| [49] | Manavgat cable-stayed bridge FE model | SVM, THT | ACC = 95.43% | HHT and THT were used to extract damage-sensitive features to train an SVM damage detection model | 2018 |

### 3.4. Hilbert–Huang Transform

Hilbert–Huang transform (HHT) was designed to represent a non-stationary signal in time and frequency. Intrinsic mode function (IMF) is used in combination with the Hilbert algorithm to locate signal frequency information over a specific time of a given signal. Like EMD, HHT is adaptive for a given signal, and it does not require a prior selection of a function or parameter to compute it. The amplitude and instantaneous phase of a signal at a specific time can be obtained by applying HHT for system identification (e.g., estimating natural frequencies and damping) and damage identification purposes by detecting structural response changes.

Several applications of HHT in damage identification, anomaly detection and system identification in different structures can be found in [50]. Despite the effectivity shown in HHT-proposed SHM systems, better performance can be achieved when HHT is combined with ML algorithms and data fusion techniques. Moreover, some drawbacks of applying HHT in SHM systems can be mitigated by combining other signal decomposition techniques and data-based models, such as ML models. Table 5 shows some proposals found in the literature that combine HHT and ML algorithms in the damage identification applications of structures.

**Table 5.** HHT review.

| Publication | Structure | Machine-Learning Algorithm | Accuracy | Purpose | Year |
|---|---|---|---|---|---|
| [51] | IASC–ASCE SHM benchmark four-story steel structure FE model | CNN | ACC = 92.36% (25 db SNR) | HHT was applied to structural response data to obtain time–frequency graphs and the marginal spectrums for CNN model training | 2021 |
| [52] | Five-story offshore platform experimental model | HHT, MEEMD, SVM | ACC = 62.5%, MSE = 0.90 | HHT and EMD were applied to obtain damage-sensitive features from vibration signals and then used to train an SVM damage detection model | 2021 |
| [47] | Fourteen-bay steel truss bridge experimental model | CEEMD, HHT, WGN, FFMLP | MSE = $2.65 \times 10^{-7}$ | A combination of CEEMDAN and HHT was proposed to extract four key damage-sensitive features to train an ANN model for damage detection, localization and severity estimation | 2020 |
| [53] | Three-story steel moment-resisting frame FE model | EMD, ANN | ACC = 98.8% | A combination of CEEMDAN and HHT was proposed to estimate first mode shapes and structural natural frequencies and compare them against reference values to determine the presence of damage | 2019 |
| [54] | 12-story-reinforced concrete frame experimental model | RBFNN | ACC = 100% | HHT was studied for structural modal parameter identification and damage diagnosis | 2014 |

## 4. SHM System Models

### 4.1. Physics-Based SHM Systems

In the case of physical asset monitoring, there are two main branches of modeling: physics-based modeling, also known as physical-law modeling, and data-driven modeling. Physics-based modeling aims to describe phenomena by formulating mathematical models that integrate interdisciplinary knowledge to generate models that replicate observed behavior. Models are commonly presented in differential equations whose complexity increases as more factors become involved. Several terms and parameters must be defined to fully describe the system phenomena. Initial and boundary conditions must be identified

to obtain physics equation solutions, and the computational cost associated with this operation can be very time-consuming for complex phenomena.

In SHM systems, these models are implemented to assess the condition of an asset under operating conditions to monitor changes that may indicate the presence of damage and shorten the remaining useful life of the asset. Finite element modeling (FEM) software implements well-known analyses, such as modal analysis, and allows the simulation of different structures and initial and boundary conditions straightforwardly. FEM software includes physical law models integrated into software libraries to perform damage analysis on a virtualized model of structures efficiently.

The complexity of modeling building structures under seismic excitation is caused by the intervening factors that can modify the behavior of these structures and by the difficulty of correctly defining their physical properties. Several works focus on estimating model parameters and uncertainties to improve the model of the structure. For example, Xu et al. [55] estimated the parameters of the structures based on linear and nonlinear regression analyses. These structures' linear and nonlinear parameters, such as elastic stiffness and yield displacement, are obtained for a three-story structure. Gomes et al. [56] addressed an inverse identification problem using numerical models and a genetic algorithm.

Model parameters obtained from experimental and recorded data can increase the model's accuracy compared to the actual measured results. However, environmental and operating conditions do not remain constant throughout the life of the structure. In addition, physical law models have drawbacks that limit their applications in some SHM systems. The time to solve the equations in a real-time SHM system impacts the response time to ensure safety and the reduction of economic losses in response systems for seismic protocols and evacuation procedures. In order to reduce the time costs of performing calculations, optimization algorithms applied to the problem of SHM are encouraged and proposed in the literature [57].

Table 6 lists physics-based proposals for SHM systems in buildings under vibration excitation for multi-story structures. In this table, two types of proposal contributions are shown: the identification of system parameters and damage. According to Farrar's classification, the level of damage identification for the practical proposals is also presented.

**Table 6.** N-story building structure health monitoring systems based on physical model techniques.

| Publication | Structure | Damage Indicator | Algorithm or Analysis Method | Damage Identification and Level | | Year |
|---|---|---|---|---|---|---|
| [58] | An eight-story physical building model in FEM software | Stiffness reduction | Vibration-based damage methods | Damage detection and localization | II | 2018 |
| [59] | A 14-story physical building prototype under vibration table | Modal frequencies | Operational modal analysis | Modal identification | N/A | 2017 |
| [60] | A five-story physical building prototype under vibration table | Stiffness reduction | Novel damage localization algorithm based on wave propagation | Damage detection and localization | II | 2020 |
| [61] | A 51-story building with accelerometers and tilt sensors | Modal frequencies | Modal parameters estimation through Bayesian algorithm combined with FFT | Modal identification | N/A | 2019 |
| [62] | A 12-story frame structure | Stiffness reduction | Hysteresis loop analysis method | Damage detection | I | 2017 |
| [63] | An 86-story physical building in FEM software | Modal frequencies | Wave-based damage detection based on propagation analysis | Modal identification | N/A | 2018 |
| [64] | A three-story frame structure | Inter-story displacement | Two novel damage indices based on the displacement of the structure | Damage detection and localization | II | 2015 |

### 4.2. Data-Based SHM Systems

Machine-learning techniques are a subset of the field of artificial intelligence. Due to their statistical nature, their vision is to address problems of interest in pattern recognition

identification and classification tasks. SHM from the ML point of view is a classification problem in which at least two states are compared in SHM systems employing ML techniques: damaged and undamaged states.

A machine-learning model extracts information in the form of features from a given data set and classifies those data features. These data-driven models require large amounts of information to train the model and avoid the overfitting problem. The generalization problem depends on the amount of available data and significant diversity of this training data to avoid overfitting and ensure a reasonable level of generalization. In an idealized situation, the data set should include the samples of the possible range of excitation that can be applied to the structure. In addition, data quality improvement using signal processing techniques, such as normalization and noise filtering, is desirable for the generation of the data set. ML algorithms are applied in the damage identification process and in analyzing the anomalous data obtained from the sensors [8,65], thereby improving data quality. In addition, signal processing techniques, such as WT and HHT also improve data quality and are applied in SHM systems [9,66].

ML techniques can be divided into supervised learning (for regression and classification tasks), unsupervised learning (anomaly detection and clustering) and reinforcement learning. The most popular ML techniques implemented in the construction of SHM solutions are support vector machines (SVMs) and convolutional neural networks (CNNs) [10,67]. In the case of neural network techniques, damage-sensitive feature selection plays a crucial role in the performance of the SHM system. SVM optimally classifies features in linear and nonlinear problems. CNN, a subset of neural network (NN) methods, include convolution operations in the hidden layer of neural networks to classify data, usually in image format. Other techniques, such as PCA, improve the features of the training data set by making them uncorrelated.

The selection of a ML technique is guided by the limitations of each technique and the requirements of SHM in terms of damage identification level and operating conditions. Identifying the modal systems of building structures can also be performed using deep neuronal networks (DNNs) [11,68]. Table 7 summarizes the ML-based proposals for SHM systems in buildings under vibration excitation for multi-story structures.

**Table 7.** N-story building structure health monitoring systems based on machine-learning techniques.

| Publication | Structure | Data Type Used for Training | Machine-Learning Technique | Damage Identification and Level | | Year |
|---|---|---|---|---|---|---|
| [69] | A four-story physical building prototype under vibration table | Acceleration response data from a physical prototype | ANN | Damage existence and localization | II | 2016 |
| [70] | An eight-story physical building mathematical model | Artificially generated dataset from an algorithm | FCN | Damage existence and localization | II | 2020 |
| [71] | A three-story physical building simulated model | Simulation-generated dataset from OpenSeesMD software | ANN | Damage detection and localization | II | 2017 |
| [72] | 30 buildings including 3, 5 and 7 stories with different structural parameters | Simulation-generated dataset from Raumoko3D software | ANN | Damage detection and localization | II | 2017 |
| [73] | An instrumented main steel frame | Experimental simulation from a physical prototype with modal shaker excitation | CNN | Damage detection and localization | II | 2016 |
| [74] | A three-story physical building-simulated model | Simulation-generated dataset from OpenSeesMD software | SVM | Damage existence, localization and severity | III | 2019 |
| [75] | A three-story steel frame structure | Intensity-based features | SVM | Damage detection | I | 2019 |
| [76] | A seven-story steel structure | Simulation-generated dataset | ANN | Damage existence, localization and severity | III | 2018 |
| [77] | A five-story steel structure | Simulation-generated dataset | ANN | Damage detection and localization | II | 2008 |

### 4.2.1. Artificial Neural Networks (ANNs)

Inspired by the working of brain neurons, artificial neural networks are the most well-known ML algorithm applied for classification and regression problems. ANN architectures consist of an input layer, several hidden layers and an output layer. Several hidden layers

can be stacked between the input and output layer according to the complexity of the ANN. Figure 2 shows a typical architecture of a forward neural network, showing input neurons, successive hidden layers and output neurons. In general, ANN power resides in the interconnection of its neurons and a set of weights associated with each interconnection. ANN learning algorithms update network weights to minimize an error function. BP-ANN is a widely used ANN where the backpropagation algorithm (BP) performs the network's training.

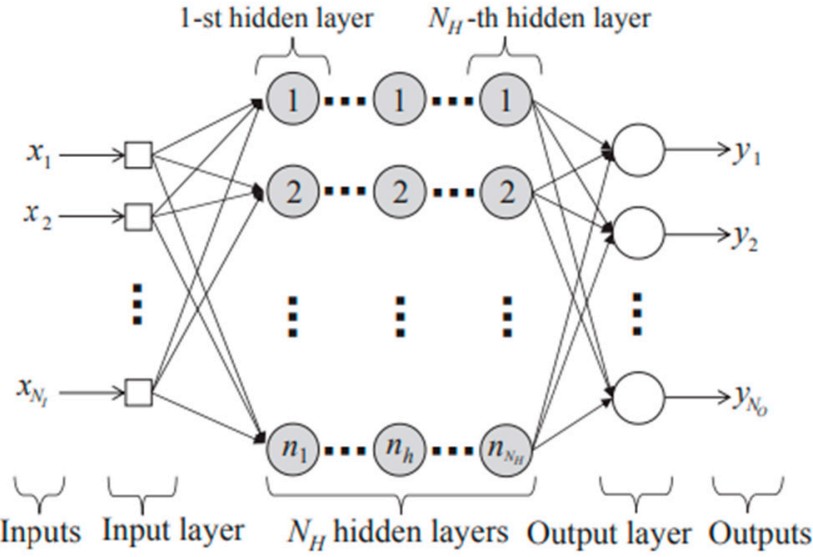

**Figure 2.** Multi-layer hidden layer feedforward ANN [78].

In many SHM systems, FE structural models generate training datasets for ANN weight calculation. In these datasets, damaged and undamaged training examples are provided. Natural frequencies and changes in the structural properties are associated with structural damage. Table 8 groups several SHM proposed systems using ANN and signal processing techniques and modal analysis for damage detection, localization and severity estimation.

Tan et al. [79] proposed a method in which the modal strain energy was used as a damage-sensitive feature and an ANN to locate and quantify structural damage in a series of damaged scenarios. A structural damage identification method using RFR functions in combination with PCA was described by Padil et al. [20] to reduce the input data size and minimize the effects of measurement uncertainties and interferences to reduce the model error. Mousavi et al. [47] proposed a damage identification methodology using the CEEDAM technique to extract damage-sensitive features from the measured data and employing an ANN to perform the classification task in various damage scenarios. The bridge damage presence, severity and location were obtained using the methodology mentioned above.

In [80], Finotti et al. presented a comparison between ANN and SVM techniques in damage identification capability using data from ten statistical indicators from acceleration measurements. These indicators were proposed to avoid manual selection and calculation of modal structural parameters that may introduce undesirable inaccuracies or errors in damage identification performance. Chang et al. [76] proposed a post-seismic evaluation method based on an ANN. This ANN was trained with simulation-generated data from a previously constructed FE model with structural parameters identified by the stochastic identification of the system subspace. Although the accuracy of damage identification was based on the identification of system parameters, it was possible to efficiently locate damage scenarios in one or several elements of the structure.

In the study conducted by Morfidis and Kostinakis [72], combinations of 14 seismic parameters together with an MLP were explored for structural damage prediction. Their

analysis observed that with at least five seismic parameters combined with an MLP network, the seismic damage state for an R/C building can be accurately predicted. Correlations between seismic parameters and MLP network training algorithms were also observed and analyzed. Using natural frequencies and modal shapes as input to the network, Padil et al. [81] defined a damage identification system using an ANN. A relationship between input and output parameters was established by defining a probability of damage existence to classify damage states. Numerical and experimental samples were used to validate the proposed method regarding modeling errors and noise interference.

Jin et al. [82] proposed an artificial neural network with an extended Kalman filter to remove temperature effects in SHM systems. This EKF was also used to establish confidence variation intervals and thresholds for structural changes that were suggestive of damage. It was shown that temperature changes induced changes in natural frequencies that can lead to incorrect damage identification if these temperature effects are not considered in the training samples. Smarsly et al. [69] provided a decentralized sensor fault detection system consisting of a set of ANNs that perform anomaly detection in an SHM wireless sensor network. An anomaly detection ANN was incorporated at each sensor node to perform local anomaly detection. Each ANN utilized sensor data redundancy throughout the sensor network using the correlation of sensor measurements in structural response.

Kourehli [83] designed an ANN-based SHM system using incomplete FEM modal data and natural frequencies for damage location and damage severity estimation using only the first two natural frequencies. Using stiffness reduction as a damage indicator, damage identification was successfully achieved in an 8-DOF spring–mass system, a supported beam, and a three-story flat frame. Natural frequencies are commonly used as damage-sensitive features in SHM systems, and the importance of eliminating or reducing interference from temperature changes was noted by Gu et al. [29]. Using a multilayer ANN, they discriminated between changes in natural frequencies due to structural damage and changes in these natural frequencies due to temperature effects. A damage index based on Euclidean distance was employed to quantify the difference between an undamaged structural state under temperature variations and a possible structurally damaged state.

Ng [78] proposed an ANN for structural damage localization and severity estimation using a Bayesian class selection method, improving the selection of optimal ANN hyperparameters (optimal transfer function and the number of neurons). Validation tests were performed using the Phase II IASC-ASCE SHM reference structure data, where ANN estimates stiffness reduction values as an indicator of damage location and severity for six damage cases. Xie et al. [84] presented a bridge damage management system using a BP-ANN algorithm that utilizes four hundred sensors randomly distributed over the bridge. A comparative study was performed employing the proposed BP-ANN algorithm, an SVM, a decision tree (DT) and a logistic regression model, showing that ANN achieves the highest accuracy in damage identification for the proposed SHM system, even under noisy operating conditions. Goh et al. [85] designed a two-step damage identification method. In the first step, a prediction of the modal shape of the unmeasured structure was performed using an ANN. The predicted modal shape was compared with a cubic spline interpolation prediction method to verify the prediction accuracy. Next, a second ANN trained with limited structural response measurement points was used to perform structural damage localization and damage severity. Shu et al. [86] proposed another ANN-BP for bridge damage identification using the statistical parameters of displacement and acceleration measurements as training input for ANN. This proposal showed that measurement noise negatively impacts damage identification performance and should be eliminated or minimized. For this approach, single and multiple damage cases were used for validation testing.

**Table 8.** ANN review.

| Publication | Structure | Excitation Source | Used Techniques and/or Algorithms | Number of Neurons | Number of Hidden Layers | Training Algorithm | Activation Function | Error | Dataset Size | Purpose | Year |
|---|---|---|---|---|---|---|---|---|---|---|---|
| [79] | A multiple steel girder composite bridge experimental model | Electrodynamic shaker | FFT, FRF | 4 ANN architectures: 10, 15, 20, 25 and 30 | 1 | LMBP | Sigmoid | MSE = 0.000029 (30 neurons) | 114 samples | Propose ANN architectures for damage detection and localization in structures | 2020 |
| [20] | A simply supported steel truss bridge FE model | Simulation | FRF, PCA | 20 | 1 | LMBP | Tangent-sigmoid | MSE = 0.015213 | 1200 random damage cases | Identify presence of damage considering several uncertainties | 2020 |
| [47] | A fourteen-bay steel truss bridge experimental model | Electrodynamic shaker | CEEMD, HHT, GWN | 34 | 1 | BP | Log-sigmoid | MSE = $2.65 \times 10^{-7}$ | A time series of 14,000 data points | Identify presence, location and severity of bridge damage | 2020 |
| [80] | Simply supported beam FE model and railway bridge experimental models | Simulation (beam) and operational conditions (bridge) | SSaI, ARN | 10 | 1 | LMBP | Sigmoid | DCR = 93.54% (beam), DCR = 87.09% (bridge) | 5400 samples (beam), 4500 samples (bridge) | Detect structural changes based on statistical indicators | 2019 |
| [76] | A twin-tower building experimental model | Shaking table | SSI | 140 | 2 | BP | Log-sigmoid | Error graph | 279,936 data points | Identify presence, location and severity of building damage | 2018 |
| [87] | A carbon fiber-reinforced plastic plate | Simulation | FT | 627 | 1 | BP | Sigmoid | ACC = 100% | 40 samples with 150 data points | Identify presence and type of plate damage | 2018 |
| [72] | A set of 30 reinforced concrete buildings numerical models | Simulation | 14 ground motion parameters, NTHA | 30 | 1 | LMBP, SCG | Tangent-sigmoid | MSE = 0.045 | 3900 vectors of size $18 \times 1$ | Propose damage-sensitive features for improving ANN structural damage prediction | 2017 |
| [81] | A single-span steel frame numerical model | Impact hammer | SRF | 20 | 1 | LMBP | Tangent-sigmoid | MSE = 0.0026 | 2400 samples | Identify presence of structural damage using noisy ANN training data | 2017 |
| [82] | A Meriden bridge FE model | Simulation | EKF | 6 | 1 | BP | Tangent | MAE = 0.0572 | 6480 samples | Propose a damage detection method using ANN and EKF in structures under temperature changes | 2016 |
| [88] | An instrumented aircraft panel and an instrumented wind turbine | PZT burst signal and operational conditions | Ensemble classifier | 1 (panel ANN) and 27 (turbine ANN) | 1 (panel), 2 (turbine) | Not specified | Not specified | MSE = 0.0098 (panel), MSE = 0.000194 (turbine) | 110 data samples (aircraft) and 3450 data samples (turbine) | Propose an ensemble design method for ANN hyperparameter selection | 2016 |

**Table 8.** *Cont.*

| Publication | Structure | Excitation Source | Used Techniques and/or Algorithms | Number of Neurons | Number of Hidden Layers | Training Algorithm | Activation Function | Error | Dataset Size | Purpose | Year |
|---|---|---|---|---|---|---|---|---|---|---|---|
| [69] | A four-story frame structure experimental model | Not specified | Cooley–Tukey FFT algorithm | 24 | 1 | BP | Not specified | RMS = 0.807 | 100 samples | Propose a detection method for sensor faults in response data of SHM systems | 2016 |
| [83] | 3 numerical models: a simply supported beam, three-story plane frame and an 8-DOF spring–mass system | Simulation | ARN | 6 ANN architectures: 30, 40, 46, 53, 44 and 51 | 1 | LMBP, GDM | Tangent | MSE = $3.44 \times 10^{-3}$ (beam), MSE = $4.77 \times 10^{-3}$ (plane frame), MSE = $9.12 \times 10^{-4}$ (spring–mass system) | 2304 samples (beam), 2592 samples (three-story frame) and 2187 samples (DOF spring system) | Propose a structural damage detection ANN using natural frequencies and incomplete structure mode shapes | 2015 |
| [29] | A simply supported steel beam girder FE model | Simulation and a piston | PCA, GWN | 2 ANN architectures: 35 and 17 | 1 | LMBP | Tangent-sigmoid | MSE = 0.1137 | 4000 samples | Use of an ANN to associate changes in modal frequencies to structural damage due temperature changes | 2015 |
| [78] | The Phase II IASC-ASCE SHM benchmark structure FE model | Simulation | Bayesian ANN design algorithm | 17 | 1 | Not specified | Tangent-sigmoid | ACC = 97% | 226 samples | A Bayesian model class selection method was proposed to select optimal ANN hyperparameters | 2014 |
| [84] | An instrumented bridge | Operational conditions | N/A | 674 | 1 | BP | Sigmoid | ACC = 98% | 8000 samples | Propose a damage detection method that achieves high accuracy and more robust performance in the presence of noise | 2013 |
| [85] | A two-span-reinforced concrete slab FE model | Simulation | SRF | Not specified | 1 | SCG | Tangent-sigmoid | MSE = 0.00342 | 3000 samples | Propose an ANN to predict unmeasured mode shape values for damage detection | 2013 |
| [86] | A Banafjäl Bridge FE model | Simulation | GWN | 4 ANN architectures: 19 and 23 (single damage case); 19 and 25 (multiple damage case) | 2 | BP | Not specified | ACC = 79% | 416 samples (single damage case) and 900 samples (multiple damage case) | Propose a BP-ANN for detection, localization and extent of structural damage using statistical indicators | 2013 |
| [35] | A 10-story low-rise building FE model | Simulation | FRF, PCA | 2 ANN architectures: 25 and 37 | 2 | BP | Log-sigmoid | ACC = 98.77% | 240 samples | Propose an ANN for detection, localization and extent of structural damage using RF as damage index and PCA | 2013 |

**Table 8.** *Cont.*

| Publication | Structure | Excitation Source | Used Techniques and/or Algorithms | Number of Neurons | Number of Hidden Layers | Training Algorithm | Activation Function | Error | Dataset Size | Purpose | Year |
|---|---|---|---|---|---|---|---|---|---|---|---|
| [89] | An aluminum beam experimental model | Mechanical roller | CC | 36 | 2 | BP | Sigmoid | ACC = 87.7% | 250 samples | This paper reported a NN technique to select damage-sensitive frequency ranges and diagnose structural damage | 2012 |
| [90] | A building FE model | Simulation | FEMA damage index | 49 | 1 | BP | Sigmoid | MSE = $1.6 \times 10^{-10}$ | 835 samples | Propose an SHM system based on ANN to predict the building damage index | 2012 |
| [91] | A four-story building FE model | Simulation | FRF, PNN | 16 | 1 | BP | Not specified | ACC = 100% | 30 samples+J24 | Propose a PNN and a BP-NN for detection, localization and extent of structural damage using RF | 2011 |
| [45] | A four-story building FE model | Simulation | WPT, GWN, Battle–Lemarie decomposition | 2 ANN architectures: 39 (single sensor, damage extend) and 90 (multi sensor fusion, damage extend) | 1 | BP | Not specified | ACC = 90% (single sensor), ACC = 100% (multiple sensor) | 183 samples | Propose a feature fusion and a neural network model for structural damage | 2011 |

Using a combination of PCA and FRF frequency–response functions, Bandara et al. [35] extracted damage-sensitive features from the structural response data to train an ANN-based damage identification system. The damage location and severity of the structure were represented by damage index classes in the output of the proposed ANN. Several noise levels were added to the acceleration measurements to show the denoising capabilities of PCA on the ANN training data and the reduction of data size to improve training time and computational cost. Mardiyono et al. [90] presented a BP-ANN architecture as part of a post-earthquake SHM response system using the FEMA damage index framework to classify the structural condition of buildings. A complete building SHM framework was developed, including a warning system module, an intelligent module, a monitoring module, and a data acquisition module. The complete function of the framework included accelerometer data collection, data processing and feeding to ANN, damage index prediction and alert notification in case of structural compromise. Wang et al. [91] presented a building damage identification method using a BP-ANN and a probabilistic neural network (PNN). In this method, the damage location process consists of three stages in which the damage location is refined at each stage. In the three stages, the damage was detected in each history, the extent of damage in the history was predicted and the compromised structural members were identified, respectively. Accuracy results indicated that the BP-ANN performed better in estimating the severity of structural damage, and the PNN suggested the damage location more accurately. Liu et al. [45] proposed structural damage identification where energy components were extracted from acceleration measurements and fused with data fusion techniques. An ANN classifier was used to identify, locate and estimate structural damage. The combination of WTERP and data fusion techniques improved the accuracy of ANN damage identification, and WPT-based component energies were shown to be an effective damage-sensitive feature.

### 4.2.2. Convolutional Neural Networks (CNNs)

Convolutional neural networks (CNNs) are maybe the most representative deep algorithm and have been adopted in several application fields of data-based classification and regression problems. A CNN is composed, essentially, of three types of layers according to its function: convolutional layers, pooling layers and fully connected layers. Convolutional layers automatically extract features from the input data through a convolution between input data and a user-defined matrix named filter. Pooling layers reduce the data size, and fully connected layers perform data classification tasks. Several layers can be stacked in the network design architecture, thereby increasing network complexity at the cost of incrementing computation training time and the amount of resources used.

CNN networks' inputs consist of n-dimensional training samples, where n can be from 1 to N. Figure 3 shows a convolutional network with an input vector of dimension 1D ($128 \times 1$) for a multiclass classification task. These inputs are, in most cases, images according to the classification or regression problem. In structural damage identification, proposals using CNN architectures use 1D acceleration records or 2D data that consist of images or matrices using reshaping methods, signal processing techniques or feature extraction methods. One advantage of using CNN models in SHM systems is that convolutional layers perform the selection of damage-sensitive features automatically.

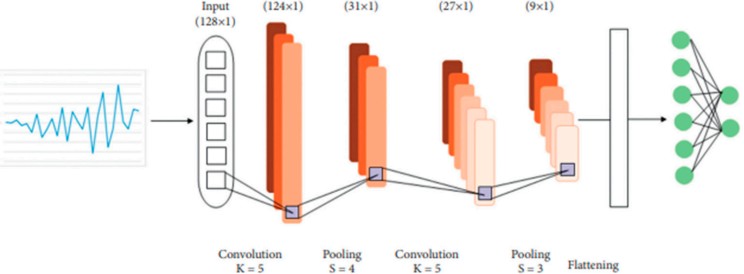

**Figure 3.** A CNN network example using raw acceleration input data [92]. Input data can be reshaped in 1D or higher dimension vector fashion.

Table 9 lists several proposals for SHM applications that employ CNNs. In these proposals, structural responses are captured as acceleration measurements are used. A complete and exhaustive review of SHM systems using CNN and mainly image data can be consulted in [93].

Some relevant limitations of the applications of CNN arise in SHM systems for structural assessment. A large amount of training data are usually required in CNN training to achieve the generalization property of the model, and the structural data of damage states are unavailable in most cases. Moreover, the structural data of damage states in large civil structures are uncommon. A possible solution is to generate damage structural state data from FE models, which is the usual approach in several SHM proposals, as shown in Table 6 However, FE-generated data rely on the accuracy of the FE models, and this also depends on the correct value of the structural parameters that are used for the model. Some of these parameter values are unknown or change over time, so they should be estimated or calculated from experimental data.

Wang et al. [51] proposed a structural damage identification method using a time–frequency plot of the acceleration signal. The marginal spectrum was used as the input to a CNN, and the hyperparameters of this network were optimized using PSO, thereby achieving 10% better accuracy than a CNN without PSO. Oh and Kim [94] explored the application of two objective function techniques for the optimal hyperparameter selection of a structural damage identification system using CNN. Their results showed a computational cost reduction of 40%. Sony et al. [95] designed a 1D-CNN to perform multiclass damage identification using bridge vibration data. The random search technique performed hyperparameter tuning to obtain the optimal network architecture. Oh and Kim [96] also proposed a hyperparameter search technique in CNN. These hyperparameters included the subsampling size and the number and size of kernels using a genetic algorithm and a multi-objective optimization technique.

Damage-sensitive features were extracted from the automatic signal using a CNN architecture in the damage identification strategy presented by Rosafalco et al. [70], which obtained good performance in damage identification and localization. In [97] Zhang et al. proposed a physics-guided CNN architecture for structural response modeling. Structural physics provide a set of constraints for CNN training, which avoid overfitting problems and reduce the size of the training dataset. de Rezende et al. [98] presented a 1D-CNN network with an electromechanical impedance-based methodology to achieve damage identification under various environmental temperature conditions, proposing a robust structural damage identification scheme against temperature effects. Sarawgi et al. [99] designed a distributed SHM damage identification system using 1D-CNN and a four-step methodology. In the first step, a multiclass classification of the vibration data is performed, and the second step consists of training the CNN with patterns of damaged and undamaged data. The damage location is identified in the third step. The first three steps are performed on each node, and in the final step, a combined data set from all nodes are used to train a global damage identification model.

Focusing on joint damage identification, Sharma and Sen [100] proposed a joint damage identification method based on using a 1D-CNN with robustness to signal noise. Single and multiple damage cases were successfully classified. Using the modal shapes and curvature differences as input to the CNN, Zhong et al. [101] proposed a damage identification system based on CNN, detecting that the modal shapes contain damage-sensitive features, which are sensitive to all degrees of damage, thereby allowing high detection accuracy. Additionally, it was shown that the parameters of the first-order modal signal have helpful information for the localization of structural damage. Seventekidis et al. [102] presented the idea of building an optimal FE model to generate training data for a CNN. A comparative case was analyzed comparing this generated optimal data set against a FE data set, showing that the latter is unreliable for CNN training given model uncertainties. This optimal FE model was obtained by initial structure measurements and, subsequently, the use of the covariance matrix adaptation evolution strategy.



Ibrahim et al. [2] evaluated several machine-learning approaches using low-cost accelerometers and noise filtering techniques for structural evaluation after excitation events, including SVM, k-nearest neighbors and CNN. This work demonstrated that the CNN-based detection method in this structure outperformed both SVM and the k-nearest neighbors' method using noise-contaminated raw measurements.

In a study by Duan and Zhu [103], the Fourier amplitude spectrum was used with a CNN to perform damage state structure identification. This work showed that the use of FAS outperforms traditional CNN proposals using time–history acceleration responses and that the proposed FAS-CNN SHM system was robust against noise effects. Gulgec et al. [104] proposed a damage identification and location method that used a CNN architecture and was trained by analytical simulations. The presented method consisted of two steps for damage identification: the first to detect the presence of damage as a binary classification problem and the second as a regression problem to estimate the damage location by predicting the damage boundaries in the structure. Using the stiffness histogram of the measured structural response and stiffness degradation as an indicator of damage, Zhou et al. [105] proposed a deep learning network to provide a real-time alarm in case of structure compromise. Zhang and Wang [106] proposed a 1D-CNN for structural state identification using raw time-domain data. GANs were used to generate training data and improve the robustness and effectiveness of structural damage identification.

To avoid the manual selection and elaboration of damage-sensitive features from the raw measurements, Yu et al. [107] proposed a CNN architecture to identify and localize structural damage. The input to the proposed network consisted of frequency-domain features obtained by FFT on the raw measurement signals. Compared with other ML techniques, including general regression neural network (GRNN) and ANFIS, the proposed network outperformed these in error classification. Tang et al. [108] proposed a structural damage identification system where real-world anomaly detection acceleration data are transformed into dual-channel time–frequency images. The anomaly detection patterns were correctly identified with an overall average accuracy of 93.5%. They highlight the importance and effect of balancing on the training dataset. Wu and Jahanshahi [109] presented a structural response prediction method using a DCNN in three structures: a linear SDOF, a nonlinear SDOF, and a three-story MDOF steel frame. This approach was compared with an MLP network regarding noise signal robustness, and the convolutional kernel analysis showed a frequency signature feature extraction conducted by the convolutional layers.

Khodabandehlou et al. [110] presented a novel approach using 2D convolutional networks from the raw measurements of a scaled bridge model. This network was able to classify the bridge's structural condition into four severe damage states, achieving 100% accuracy for the validation test. Avci et al. [111] presented a decentralized 1D-CNN architecture for structural damage identification. In this SHM system, each sensor performs damage identification locally using a 1D-CNN per sensor. Damage identification and localization were achieved through the classification network of each sensor, and the need for data transmission and aggregation is reduced because each sensor analyzes its local measurement data. Azimi and Pekcan [112] designed a CNN SHM system for damage identification and localization using compressed acceleration response histograms as network input. Deep transfer learning techniques were applied to this structural damage classification problem and proved feasible for damage identification in similar structures using discrete acceleration histograms.

**Table 9.** CNN review.

| Publication | Structure | Excitation Source | Signal Preprocessing Techniques | Dataset Size | Convolution Layers Size | Pooling Layers Size | Activation Function | Accuracy | Purpose | Year |
|---|---|---|---|---|---|---|---|---|---|---|
| [51] | The Phase II IASC-ASCE SHM benchmark structure FE model | Simulation | HHT | 1000 samples of size 16 × 4000 | CNN 1D: 10 × 1, 10 × 1, 6 × 1 (3 layers), CNN 2D: 14 × 14, 5 × 5 (2 layers) | CNN 1D: 3 × 1, 6 × 1 (2 layers), CNN 2D: 2 × 2, 3 × 3 (2 layers) | Softmax | ACC = 100% | Propose a novel damage identification method using HHT and CNN with improved accuracy in comparison with CNN and SVM methods | 2021 |
| [94] | A steel beam numerical model | Numerical model forces | GWN | 2000 samples of size 10 × 10 | CNN 1: 5 × 1, 3 × 1 (2 layers), CNN 2: 4 × 4, 2 ×2 (2 layers) | CNN 1: 4 × 1, 2 × 1 (2 layers), CNN 2: 2 × 2, 1 × 1 (2 layers) | Sigmoid | RMS = 2.5 | Propose a method for CNN hyperparameters selection in SHM systems | 2021 |
| [95] | Z24 bridge experimental model | Electrodynamic shaker | Statistical scaling | 1,231 time series with 65,530 data points | 16 × 16 (1 layer) | N/A | Softmax | ACC = 0.85 (pier settlement), ACC = 0.66 (tendon rupture) | Propose a 1D-CNN for multiclass structural damage detection using limited datasets | 2021 |
| [96] | A three-story building frame experimental model | Shaking table | N/A | 2000 samples of size 10 × 10 | 8 × 8, 8 × 8, 7 × 7 (3 layers) | 8, 8, 7 (3 layers) | Sigmoid | RMS ≈ 5.6 | Propose a method for CNN hyperparameters selection in SHM systems | 2021 |
| [70] | An eight-story building numerical model | Lateral and vertical loads applied at each story | GWN | 9216 samples | 8 × 1, 5 × 1, 3 × 1 (3 layers) | N/A | ReLU and Softmax | ACC = 99.3% | Propose an FCN architecture for damage detection and localization in structures | 2020 |
| [97] | A six-story hotel instrumented building | Historically recorded response | Butterworth HPF | 11 samples with 7200 data points | 4 × 1, 4 × 1 (2 layers) | N/A | ReLU | Confidence = 93% (worst case) | Propose a CNN architecture to develop a surrogate model for modeling the seismic response of building structures | 2020 |
| [98] | A three-aluminum-beam experimental model | Mass structure addition and temperature changes | N/A | 900 samples with 888 data points | 237 × 1 (1 layer) | 2 × 1 (1 layer) | ReLU and Softmax | ACC = 97% | Propose a combination of CNN and EMI methods for structural damage prediction | 2020 |
| [99] | The Phase I IASC-ASCE SHM benchmark structure FE model | Simulation | N/A | 6 samples of size 1000 × 400 | 41 × 1, 41 × 1 (2 layers) | 41 × 1, 41 × 1 (2 layers) | tanH | ACC = 96.11% | Propose a 1D-CNN for predicting damage from vibration data of structures | 2020 |
| [100] | A three-story shear frame FE model | Simulation | GWN | 19,800 samples | 128 × 1, 64 × 1, 32 × 1, and 16 × 1 (4 layers) | 2 × 1, 2 × 1, 2 × 1 and 2 × 1 (4 layers) | Softmax | ACC > 81.8% | Propose CNN-based approach to identify and locate damage in structural joints | 2020 |

**Table 9.** *Cont.*

| Publication | Structure | Excitation Source | Signal Preprocessing Techniques | Dataset Size | Convolution Layers Size | Pooling Layers Size | Activation Function | Accuracy | Purpose | Year |
|---|---|---|---|---|---|---|---|---|---|---|
| [101] | A steel truss FE model | Simulation | N/A | 5326 samples of size $15 \times 3$ | $3 \times 3, 2 \times 2, 1 \times 1, 1 \times 1$ (4 layers) | N/A | ReLU and Softmax | ACC > 90% (50% damage degree) | Propose a CNN method to locate damaged rods using first-order mode shapes and mode curvature differences | 2020 |
| [102] | A linear steel beam FE model | Electrodynamical shaker | N/A | 7500 samples | $12 \times 12, 10 \times 10, 8 \times 8$ (3 layers) | $4 \times 4, 5 \times 5, 6 \times 6$ (layers) | ReLU | 91.75% and 92.18% | Present a novel SHM damage detection method based on a CNN trained with FE and experimental data | 2020 |
| [2] | Four- and eight-story building FE models | Simulation | HPF | 5000 samples | $50 \times 50, 10 \times 10$ (2 layers) | $20 \times 20, 2 \times 2$ (2 layers) | ReLU and Softmax | ACC > 90% (several noise levels) | Propose a CNN method using raw data for structural damage detection | 2019 |
| [103] | A tied-arch bridge FE model | Simulation | FDD, FFT | 10,000 samples | $3 \times 7$ (2 layers) | $4 \times 1$ (1 layer) | ReLU and Sigmoid | ACC = 99.92 (worst case) | Propose a method for structural damage identification using Fourier amplitude spectra of bridge data | 2019 |
| [104] | A structural connection FE model | Simulation | GWN | 60,000 samples | $3 \times 3$ (3 layers) | $2 \times 2$ (2 layers) | tanH | ACC = 92.5% (6 cm crack), ACC = 95% (8 cm crack) | Propose a CNN for damage detection and localization in structures | 2019 |
| [105] | A three-story full-scale building experimental model | Shaking table | Bouc–Wen model, Baber–Noori hysteretic model | 35,400 samples | $10 \times 1$ (1 layer) | N/A | Sigmoid | ACC = 97.2% | Propose a DLN to predict the damage index (DI) of stiffness degradation for structures during earthquakes | 2019 |
| [106] | A steel Warren truss bridge scale experimental model | Impact hammer | N/A | Not specified | $7 \times 1, 5 \times 1, 3 \times 1$ (3 layers) | $3 \times 1, 3 \times 1, 3 \times 1$ (3 layers) | ReLU | ACC = 100% | Propose a new CNN framework to detect and locate damage in different structural crack scenarios | 2019 |
| [107] | A five-level benchmark building experimental model | Simulation | GWN, FFT, PSD | 2832 samples of length 5 | $1000 \times 1, 30 \times 3, 10 \times 3$ (3 layers) | $3 \times 1, 3 \times 1, 3 \times 1$ (3 layers) | ReLU and Softmax | RMSE = 0.0163 | Propose a novel method based on DCNN for damage identification and location in building structures | 2018 |
| [108] | A long-span cable-stayed instrumented bridge | Operational conditions | FFT | 333,792 samples of size $100 \times 100 \times 2$ | $41 \times 41 \times 2$ (1 layer) | $2 \times 2$ (1 layer) | ReLU and Softmax | ACC = 93.5% | Propose a novel method based on CNN using time series that are visualized in the time domain and frequency domain | 2018 |

**Table 9.** *Cont.*

| Publication | Structure | Excitation Source | Signal Preprocessing Techniques | Dataset Size | Convolution Layers Size | Pooling Layers Size | Activation Function | Accuracy | Purpose | Year |
|---|---|---|---|---|---|---|---|---|---|---|
| [109] | A three-story steel frame experimental model | Shaking table | N/A | 600 samples | 5 × 8 (7 layers) | N/A | ReLU and tanH | RMS = 0.41 (worst noise case) | This study presents a DCNN approach to estimate the dynamic response of structures | 2018 |
| [110] | A highway bridge experimental model | Shaking table | N/A | 48 samples of size 122 × 70 | 4 × 4, 8 × 8, 16 × 16, 32 × 32, 32 × 32 (5 layers) | 2 × 2 (5 layers) | ReLU and tanH | ACC = 98.437% | Propose a two-dimensional CNN-based SHM system to identify structural damage states | 2018 |
| [111] | A steel structure experimental model | Electrodynamic shaker | Signal normalization | 245,760 samples | 80 × 1 (2 layers) | N/A | Not specified | Error graph | Propose a CNN SHM-based approach using raw signal without preprocessing or signal extraction techniques to detect structural damage | 2018 |
| [113] | An instrumented main steel frame | Electrodynamic shaker | N/A | 6 samples of size | 42 × 1 (1 layer) | 2 × 1 (1 layer) | Not specified | Average error = 0.54%. | Propose a damage detection and localization method using an adaptive 1D-CNN | 2017 |
| [112] | The Phase II IASC-ASCE SHM benchmark structure FE model | Simulation | GWN | 262,144 samples | 3 × 1 (9 layers) | 2 × 1 (3 layers) | ReLU and Softmax | ACC = 90–100% (several damage cases) | Present a novel CNN-based approach that uses acceleration data to estimate structural responses through transfer learning (TL)-based techniques | 2016 |

4.2.3. Support Vector Machines (SVMs)

A support vector machine (SVM) is a popular machine-learning algorithm used for classification and regression problems. Its working principle is essential to maximize the distance value between a set of vectors, named support vectors, and a defined hyperplane. The maximizing goal in SVM can be seen as an optimization problem, and several approaches can be used to solve this problem. An exciting feature of the SVM decision boundary is that this boundary is the best possible boundary between a set of data classes for a given set of data and an SVM formulation. Table 10 shows that several SVM structural approaches for damage identification have high accuracy. However, SVM models' training stage is usually computationally expensive and unsuitable for real-time SHM systems that rely on post-training updates and ML model improvement. A common practice to reduce this computational cost is to combine SVM models with signal processing techniques such as PCA, WT and HHT, or damage-sensitive feature extraction methods such as RFR. Figure 4 shows an example of an SVM-based structural damage identification system.

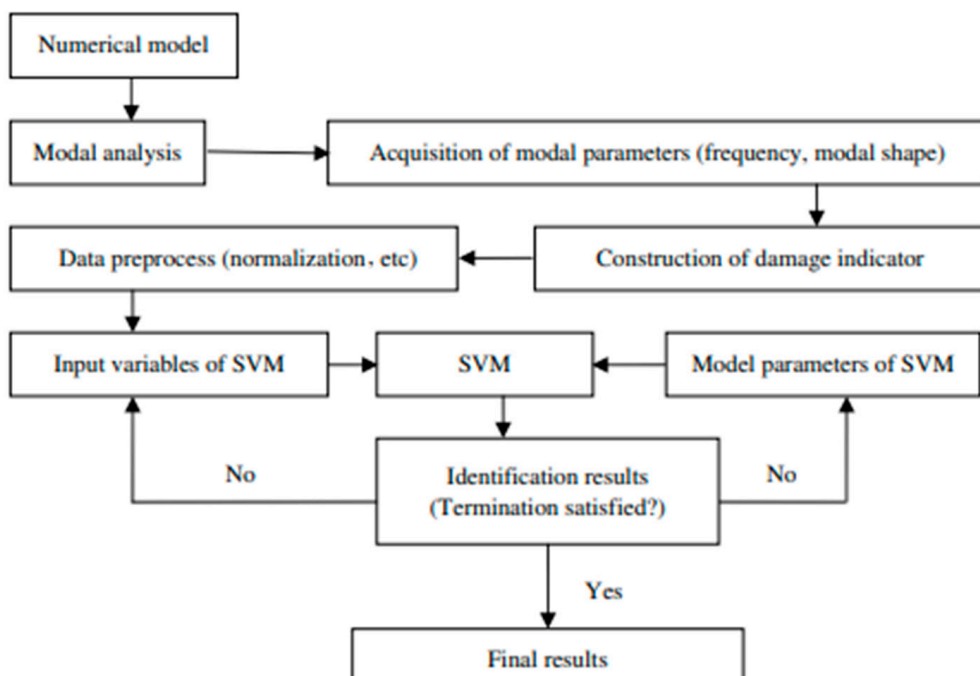

**Figure 4.** An example of an SVM-based structural damage identification system [114].

Cuong-Le et al. [115] proposed a combination of PSO and SVM for structural damage identification, location and severity. Four ML algorithms, including ANN, DNN, ANFIS and SVM, were trained and compared to predict damaged elements and damage severity. The proposed PSO-SVM combination achieved the highest accuracy among the compared techniques and correctly predicted all damage locations in the validation test. In a proposal by Agrawal and Chakraborty [116], Bayesian optimization (BO) was used in the hyperparameter search of an SVM, in which the SVM was used to perform damage identification. A comparison between BO and PSO in hyperparameter optimization search showed that BO was an efficient method that accelerated the optimization problem. Diao et al. [52] designed an SHM damage identification system, in which MEEMD and HHT were applied to acceleration measurements to extract damage characteristics through Hilbert spectrum energy. The SVM trained with Hilbert energy identified, located and estimated the severity of the damage in the structure, and the theoretical and experimental validation of the proposed method was provided.

**Table 10.** SVM review.

| Publication | Structure | Excitation Source | Signal Processing Technique | Type of Kernel | Classification Error | Dataset Size | Purpose | Year |
|---|---|---|---|---|---|---|---|---|
| [115] | Truss bridge structure FE model and two-story frame structure FE model | Simulation | PSO | GRBF | RMSE = 0.0461 (truss structure), RMSE= 0.0621 (frame structure) | 1000 samples | Propose an ML-based method to detect and locate structural damage using noisy measurements | 2021 |
| [116] | The Phase II IASC-ASCE SHM benchmark structure FE model | Simulation | PSO | GRBF | ACC = 100% | 1768 samples | Propose a method to train an SVM classifier to optimally detect structural damage | 2021 |
| [52] | A five-story offshore platform experimental model | Shaking table | HHT, MEEMD | GRBF | ACC = 62.5%, MSE = 0.90 | 50,000 data points | HHT was used to extract damage-sensitive features from raw signals in order to detect, locate and estimate the severity of structural damage | 2021 |
| [51] | 12-story building numerical model and laboratory-scaled steel bridge experimental model | Simulation and a rubber hammer | Deep autoencoder, GWN | GRBF | ACC = 97.4% (building), ACC = 91.0% (steel bridge) | 15,600 samples (building model). 5810 samples (steel bridge) | Propose an SHM system for structural damage detection using an autoencoder to achieve high accuracy in several damage cases | 2021 |
| [117] | Two planar steel frame structure numerical models of 18 elements and 49 elements | Simulation | DEA | Newly proposed kernel function | ACC = 90.05% (double damage case 49 elements) | 2500 samples | Propose a two-step method for structural damage location and extend it through the combination of SVM (a first step) and DEA (a second step) | 2020 |
| [118] | Steel frame structure experimental model and wooden house FE model | Shaking table and simulation | DNN | GRBF | ACC = 94.7% (steel frame), ACC = 90% (wooden structure) | 6600 samples (steel frame structure), 100 samples (wooden house) | An SVM-based SHM was proposed to automatically detect several damage patterns from training data to improve the damage detection generalization capabilities of an SHM system | 2020 |
| [80] | Simply supported steel beam FE model and instrumented bridge | Electrodynamic shaker | SF | GRBF | ACC = 85.87% | 19 samples of 2,400,000 data points (beam), 504 samples (bridge) | Ten statistical parameters of acceleration signals are used as damage-sensitive features to train an SHM system | 2019 |

**Table 10.** *Cont.*

| Publication | Structure | Excitation Source | Signal Processing Technique | Type of Kernel | Classification Error | Dataset Size | Purpose | Year |
|---|---|---|---|---|---|---|---|---|
| [75] | 3D-reinforced concrete moment frame FE model | Simulation | Cumulative intensity measures | GRBF | ACC = 83.1% | 6240 samples | Propose a structural damage detection and localization SHM system using an SVM model | 2019 |
| [74] | 3D-reinforced concrete moment frame FE model | Simulation | GWN | GRBF | ACC = 92.48% | 5400 samples | A framework was presented for near real-time prediction of damage existence, location and severity of building damage | 2018 |
| [119] | 31-bar planar truss FE model and double-layer grid FE model | Simulation | LS-SVM | Not specified | MSE = 0.00051 (planar truss), RMSE = 0.0129 (double layer grid) | 11,625 data points | Present a comparative study of the application of several ML techniques for damage detection of structures | 2018 |
| [120] | Three-story plane steel frame FE model and 8-DOF spring–mass system experimental model | Simulation and electrodynamic shaker | LS-SVM | GRBF | MSE = $3.243 \times 10^{-7}$ (building), MSE = $7.303 \times 10^{-9}$ (plate) | 1944 samples (building), 2333 samples (plate), 2187 samples (8-DOF system) | Propose structural damage detection, localization and severity using an LS-SVM approach | 2018 |
| [41] | Offshore platform FE model and an instrumented scaled model | Simulation and shaking table | TFA, WPD, PCA | GRBF | ACC = 78.125% (FE model), ACC = 84.375% (instrumented model) | 120 samples of 10,000 data points (FE model), 48 samples of 17,500 data points (instrumented model) | WPD and PCA were used as damage-sensitive feature extractors to train an SVM model for robust structural damage detection against measurement noise | 2018 |
| [121] | Three-story frame structure experimental model | Electrodynamic shaker | GA, PSO, grid search techniques, RE | GRBF | ACC = 100% | 510 samples | Three optimization techniques were used to optimally select SVM hyperparameters using acceleration signals to perform structural damage detection tasks | 2017 |
| [122] | IASC-ASCE SHM benchmark dome truss FE model | Simulation | WPRE, HM | Newly proposed kernel function | MAE = $1.02 \times 10^{-2}$ (benchmark), MAE = $1.02 \times 10^{-2}$ (dome truss) | 162 samples of 54,000 data points | Propose a new SVM kernel in order to increase the speed and the accuracy of LS-SVM for structural damage detection | 2016 |

**Table 10.** *Cont.*

| Publication | Structure | Excitation Source | Signal Processing Technique | Type of Kernel | Classification Error | Dataset Size | Purpose | Year |
|---|---|---|---|---|---|---|---|---|
| [123] | Three numerical models: simply supported beam, two-story plane frame, 8-DOF spring–mass system | Simulation | IRS, CSA, LS-SVM | GRBF | MSE = $3.0204 \times 10^{-10}$ (beam), MSE = $7.2822 \times 10^{-4}$ (plane frame), MSE = $4.2270 \times 10^{-10}$ (8-DOF spring–mass system) | 2187 samples (beam), 1944 samples two-story plane frame), 2315 samples (8-DOF spring–mass system) | Propose a new SHM damage detection system using incomplete mode shapes and structural natural frequencies | 2016 |
| [27] | Three-story frame aluminum structure experimental model | Bumper | AANN, FA, MSD, SVD, SVDD, KPCA | GRBF | ACC = 96.64% (one class SVM) | 17 samples of 4096 data points | Four kernel SVM algorithms are tested to predict and detect structural damage in a building structure | 2016 |
| [42] | The Phase II IASC-ASCE SHM benchmark structure FE model | Simulation | WPD | GRBF | ACC = 91.75% | 99 samples | Propose a structural building damage detection system based on an SVM using data fusion techniques | 2015 |
| [124] | A supported beam FE model | Simulation | BP-NN | Not specified | ACC = 100% (single crack), ACC = 99.9% (double crack) | Not specified | Propose a structural damage detection and localization method combining an SVM and BP-NN in beam structures | 2014 |
| [32] | Building structure experimental model and Sydney Harbour-instrumented bridge | Electrodynamic shaker and operational conditions | PCA, RP, PAA | GRBF | ACC = 97% (supervised SVM), ACC = 71% (unsupervised SVM) | 3240 samples of 8192 data points (building), 6370 samples of 3600 data points (bridge) | Data dimensionality reduction techniques were used to generate training datasets for an SVM model for structural damage detection in buildings and bridge structures | 2014 |
| [43] | The Phase II IASC-ASCE SHM benchmark structure FE model | Simulation | LS-SVM, PSO, HSA, SF, WPT | GRBF | ACC = 100% (PSHS), ACC = 98.2% (Harmony), ACC = 98.68% (PSO) | 16 samples of 40,000 data points | Present a comparative study between an ANN model and LS-SVM model using hyperparameter optimization techniques to achieve optimal structural damage detection | 2014 |
| [125] | Five-story, nine-story and twenty-one-story shear structures FE models and five-story steel experimental model | Simulation and electrodynamic shaker | Damage location indicators | Not specified | ACC = 100% | Not specified | Propose a structural damage detection and localization SHM system using an SVM model | 2013 |

**Table 10.** *Cont.*

| Publication | Structure | Excitation Source | Signal Processing Technique | Type of Kernel | Classification Error | Dataset Size | Purpose | Year |
|---|---|---|---|---|---|---|---|---|
| [126] | Three-story building structure experimental model | Magnetorheological dampers | ARM, WARM, DEF | Linear kernel and GRBF | RMSE = 0.0380 (AR, damaged system), RMSE = 0.0549 (WAR, damaged system) | 100 samples | Propose SHM system using a combination of ARM and WARM to extract damage features from acceleration data and an SVM for damage detection purposes | 2013 |
| [114] | Simply supported bridge numerical model | Simulation | GA | GRBF | ACC= 100% (damage localization), ACC= 98.16% (triple damaged extend) | Not specified | A GA algorithm was used to select the optimal hyperparameters for an SVM structural damage classification model | 2011 |

Seyedpoor and Nopour [117] proposed a two-step method using a differential evolution algorithm (DEA) and an SVM for damage identification in a steel structure. In the first step, the location of the potential damage is determined using an SVM, and in the second step, the severity of the damage and a more accurate location is obtained using the DEA. The results obtained showed high accuracy in the capabilities of the damage identification system. Kohiyama et al. [118] proposed the integration of a DNN and SVMs for the SHM damage identification framework. SVMs were used in the output layer of the DNN to detect data associated with an unlearned structural damage pattern. The added value of the proposed framework is the correct classification of unlearned damage patterns that may lead to misclassification, thereby improving the generalization feature of the classification model for damage identification.

Sajedi and Liang [75] proposed a damage identification SVM system to determine the presence and location of damage using intensity-based features as damage-sensitive features. This SHM system achieved over 83% accuracy in damage classification.

Intensity-based damage features were also used in a system proposed by Sajedi and Liang [74], in which an SVM classifier performed the classification task. The optimal damage-sensitive features and hyperparameters of the SVM were obtained using Bayesian optimization techniques. The proposed system showed robustness against signal distortion and reliability in detecting damage to building structures.

Ghiasi et al. [119] presented a comprehensive comparative study among several ML algorithms, including BP-NN, LS-SVMs, ANFIS, RBFN, LMNN, ELM, GP, multivariate adaptive regression spline (MARS), random forests and Kriging, for damage identification classification, location and severity estimation tasks. The results obtained indicate that the Kriging and LS-SVM models better predict damage location and severity than the other ML models.

Kourehli [120] presented a two-step unmeasured mode prediction method using two LS-SVMs and limited sensor measurements. The first step estimates missing modal shapes through an LS-SVM. In the second step, when complete modal system information is available, a second LS-SVM performs structural damage identification. The effects of noise and modeling error interference were analyzed, showing that the proposed system is robust against these effects. In the damage identification proposal presented by Diao et al. [41], transmissibility functions, WPEV and PCA were calculated on the acceleration response data and fed into an SVM damage classification model. WPEV and PCA were used as damage-sensitive feature extractors to construct the training samples of the ML model. As an improvement of the proposed damage identification scheme, the training data size was greatly reduced with the proposed strategy. The SVM classifier performed damage localization, and damage severity was obtained from the SVM regression result.

Gui et al. [121] presented a comparative study of optimization techniques combined with SVM models. Grid-search, PSO, and GA were used to optimize the SVM hyperparameters in structural damage identification, and the selected damage features included selected autoregressive and residual error features. The application of the above techniques showed improved SVM prediction performance, and the GA with AR features shows the best classification results. Ghiasi and Noori [122] proposed a new method for structural damage identification using an SVM with a newly proposed kernel. A Littlewood–Paley wavelet kernel was presented to improve the accuracy of the SVM for the SHM damage classification task, and the social harmony search algorithm was used to optimize the hyperparameters of the SVM. The results improved damage identification accuracy with the proposed kernel compared to other kernels combined.

An LS-SVM-based damage identification method was proposed by Kourehli [123] using incomplete modal data and natural frequencies as training samples and an iterated improved reduction system method. Coupled simulated annealing (CSA) was used to determine the optimal LS-SVM fitting parameters combined with a 10-fold cross-validation method. The experimental validation of the method was performed by applying it to three

different structures, and the results showed an acceptable performance of the LS-SVM in damage identification.

Santos et al. [27] conducted a study in which four kernel-based ML models for damage identification tasks under various noise conditions were compared to analyze their performance. Among the kernel-based ML models, one-class support vector machine, support vector data description, kernel principal component analysis and greedy kernel principal component analysis were selected and used for damage identification. Compared with the previous studies performed by other ML models, kernel-based damage identification showed better accuracy results and better generalization ability.

Zhou et al. [42] presented an a posteriori probability support vector machine based on Dempster–Shafer evidence theory combined with a multi-sensor data fusion strategy. Signal energy features were extracted from vibration data using WPT and then processed by a posteriori probability support vector machine (PPSVM). The experimental results demonstrated that the proposed method was a robust structural damage identification method due to the fusion of information sensors and the application of the Dempster–Shafer evidence theory. Yan et al. [124] proposed a beam structural damage identification method for crack detection using a BP-ANN model and an SVM. The training data were generated through a FE beam model, and the strain differences between the models were considered damage indicators. The two ML models were used and compared, and the results indicated that both ML models performed well in crack identification for single and two-crack damage cases.

In the SHM damage identification approach presented by Khoa et al. [32], random projection (RP) was used to reduce the size of the vibration data, thereby reducing the computational processing time in structural monitoring. Data anomalies or structural damage were detected through one-class SVM and SVM models using the reduced bridge data. The results showed that the combination of SVM and RP improves the computational speed without affecting the detection accuracy.

Ghiasi et al. [43] presented a comparative study between ANN and LS-SVM for structural damage identification. The wavelet power spectrum was used for feature extraction, and the particle swarm harmony search (PSHS) algorithm was used for the hyperparameter selection of LS-SVM and ANN. The increased model performance of both ML techniques was achieved with the PSHS optimization method in terms of accuracy, and the combination of PSHS and LS-SVM was better than ANN with PSHS.

HoThu and Mita [125] designed an SVM-based method using only the first natural frequencies for damage identification and localization in shear structures as input data. The experimental results showed that the proposed system performed damage localization in a five-story laboratory model with response data from two accelerometers: one for the basement and the other from the roof of the structure. Kim et al. [126] presented a novel SHM framework for damage identification in smart structures. DWT, autoregressive models, and SVM were combined in this proposal, in which damage-sensitive feature extraction is performed from wavelet-based AR time series models. The detection results showed that the proposed system effectively detected structural damage in a three-story structure. Liu and Jiao [114] proposed a genetic algorithm to select the optimal hyperparameters of the SVM. Figure 4 shows the diagram with the proposed structural damage detection methodology based on the combination of a SVM and a GA.Mode shape and frequency ratios were used as input data for the SVM model. In the results, the combination of the GA-SVM damage identification method showed an accuracy of 98.16% for the case of single, double and triple damage elements, outperforming other methods, such as RBF networks and GA-BP networks.

4.2.4. Unsupervised Learning Algorithms

Unlike the supervised machine-learning methods discussed in the previous sections, unsupervised learning-based methods use training data samples without the need for class labels. Intuitively, these types of methods avoid the problem of defining possible multiple structural damage cases, including multiple structural damage scenarios. By applying unsupervised methods, two types of detections can be performed from the data obtained by the sensor network measurements of SHM systems: novelty detection and anomaly detection.

To address the drawbacks of supervised machine-learning approaches applied to structural damage identification, including sample labeling and the need for data from multiple damage scenarios, Wang and Cha [51] presented an unsupervised ML approach that uses an autoencoder to extract features from the raw measurement data. An SVM was trained with the extracted features, allowing it to detect even small damages (<10% stiffness reduction) in a structural member of a steel bridge.

Entezami and Shariatmadar [127] proposed two new damage indices for damage localization and quantification from time series modeling using an AR model. In this approach, thresholds are defined to detect possible structural damage without establishing damage patterns. Daneshvar and Sarmadi [128] explored an unsupervised anomaly-based method for short- and long-term SHM using an anomaly score and also setting an alarm threshold to establish damage. The core of the method implemented unsupervised feature selection through a one-class nearest neighbor rule.

de Almeida Cardoso et al. [129] proposed a damage detection framework based on outlier detection in the acquired data. The data descriptors used in this proposal included expected data value, data variability, data symmetry, and data flatness. In this proposal, a k-medoid clustering technique was used to detect outlier data using a proposed symbolic distance-voting scheme. Eltouny and Liang [130] developed a post-earthquake damage detection and localization approach that combines intensive cumulative measurements with an unsupervised learning algorithm. This approach consists of two stages: in the first stage, a feature extraction process is performed to extract the cumulative measures and in the second stage, joint probabilities are used as damage indicators obtained through the kernel multivariate maximum entropy method.

This review will emphasize supervised learning methods such as ANN, CNN and SVM in combination with preprocessing techniques such as WT, EMD, HHT and PCA. Despite the advantages of using unsupervised techniques in the problem of structural damage identification, the use of supervised techniques is the most adopted branch in the literature.

*4.3. Model Type Comparative in Building SHM Systems*

Table 11 analyzes the advantages and disadvantages of applying the physics-based model and the data-based model based on reviewing the proposals mentioned above.

Recent proposals implement hybrid approaches that improve SHM damage identification models and integrate physics-based and data-driven modeling solutions. One of the most common strategies is to build artificial datasets used to train ML models from FEM-generated data. Conversely, FEM parameters can be estimated from the output of an ML regression model and improve a structural virtual model.

**Table 11.** Advantages and disadvantages of a physics-based model and data-based models in SHM building applications.

| Model Approach | Advantages | Disadvantages |
|---|---|---|
| Physics-based SHM | <ul><li>The model parameters have a straightforward physical interpretation. Stiffness changes and displacements are consistent as damage-sensitive features.</li><li>It can reach all the levels of damage identification if the parameters are defined or estimated in the building structure.</li><li>The effect of the variation of the parameters can be estimated in the final result of the model. Parameter variation allows the simulation of different scenarios, and the structural safety thresholds can be established.</li></ul> | <ul><li>The calculations of the solution of the model equations may not be feasible for real-time SHM applications, where the complexity of the structure requires long processing times.</li><li>The uncertainties and changing parameters may reduce the accuracy of the output model. Therefore, an estimation using other methods, such as model-based techniques, is encouraged.</li></ul> |
| Data-based SHM | <ul><li>Noise and environmental effects on the data collected by the sensors can be minimized for the classification performed by the ML model.</li><li>Damage identification can be performed even if the parameters of the structure are unknown or cannot be estimated.</li></ul> | <ul><li>The solution process is hidden from the user, so the rationale for successful damage classification is not explicit.</li><li>The training data set must be large enough to avoid overfitting, especially in algorithms such as CNN. In addition, obtaining the training samples of damage states is in most cases limited to an artificially generated dataset from simulations.</li><li>Computational training times can be costly for some ML algorithms (SVM, for example). Real-time SHM monitoring systems require faster methods such as NN solutions.</li></ul> |

## 5. Uncertainties Effect Minimization in SHM Systems

Within the training and operation process of SHM systems, model uncertainties and monitoring system anomalies adversely affect the performance and the damage detection system capability. These uncertainties can be found in the data that are used for training and in the data that are processed during system operation.

In the training phase of the machine-learning model, it is common to use synthetic training data generated from FEM models of the structure of interest. The simulation of the structural response facilitates the task of generating data under different operating and structural damage conditions. The initial model of the structure of interest must accurately reflect the structural behavior and response, as it is the starting point for comparison with future states to determine the presence of structural damage. To minimize the uncertainties of the structural FEM model, several solutions have been proposed in recent years to obtain a model with reliable parameters and several solutions have been proposed for the optimal way to update FEM models in the SHM area.

Lee and Cho [131] proposed a direct FEM update strategy based on the identification of modal parameters and their comparison with the predicted structural response from measurements on a bridge structure. Jafarkhani [132] presented an evolutionary strategy in combination with a preliminary damage detection update triggering process and an ARMA model. An iterative FEM model updating process is used in Pachon's approach [133], where a sensitivity analysis determines the most significant structural parameters that have influence on the overall behavior of the structure. A survey of different FEM model updating methods in the area of SHM can be found in [134]. In the case of bridge-type

structures, an analysis of the different model updating techniques can be found in Sharry's work in [135].

On the other hand, during SHM systems operation it is necessary to identify the impact of environmental and operational effects on structural parameters to minimize their effects on damage-sensitive features, especially in these structures under harsh or extreme conditions. For example, high sensor sensitivity measurements without noise removal techniques may only be permissible when environmental and operational conditions remain unchanged and this scenario is very uncommon in the implementations of SHM systems.

Yuen and Kuok [136] conducted a study to analyze the environmental effects (temperature and humidity) on the modal frequencies of the structure. Through one year of measurements in a 22-story building, Yuen identifies a correlation between the mentioned environmental conditions and the modal frequencies of the structure. The normalization of the sensing system data can reduce the effects of environmental noise and sensor failures. Some of these effects may be correlated with damage and should be eliminated to avoid a false-positive error in damage identification.

Cross et al. [137] employed cointegration and (PCA) techniques to remove operational and environmental effects and extract damage-sensitive features. Damikoukas et al. [138] proposed an identification method based on sensor measurements with noise to obtain the stiffness and damping matrices of a structure model. A final model including these parameters obtained from the noise measurements predicts building response to seismic excitation. Using SVM, relevant vector machines (RVM) and cointegration techniques, Coletta [139] implemented an SHM to monitor the structural condition of a historic building. This SHM system only considers the identified frequencies of the given structure and not the environmental effects of performing structural monitoring. Temperature variation is another environmental factor that can impact the performance of an SHM system and its ability to accurately detect damage, as the sensors can be affected by these variations. Huang et al. [140] proposed a new damage identification system that employs a new algorithm, which includes the effect of temperature variations and their correlation with structural properties. A genetic algorithm (GA) is also used to solve the optimization of the process.

Other approaches in the area of unsupervised learning that consider the effect of uncertainties in measured data have been proposed for anomaly detection systems that are resistant to uncertainty in sensor data. Yan et al. [141] studied the use of symmetric Kullback–Leibler (SKL) distance and transmissibility function to perform structural anomaly detection by comparing a baseline state and potential damage scenarios. Bayesian inference and a Monte Carlo discordance test are used as a statistical screening scheme to deal with measurement uncertainties. To improve the probabilistic function of the anomaly detection model, Mei et al. [142]. used the Bhattacharyya distance of TF. The performance of this proposal outperforms other proposals using Mahalanobis distance-based methods and have better resistance against the effects of uncertainties. Sarmadi and Karamodin [143] presented a Mahalanobis-squared distance-based method that combines the one-class kNN rule and an adaptive distance measure. In this study, the clustering process eliminates the effects of varying environmental conditions on the anomaly detection process.

## 6. Discussion and Summary

Within the advantages and limitations of physics-based SHM systems, we summarize the following highlights when choosing this approach in the design of a structural damage identification or anomaly detection system.

### 6.1. Physics-Based SHM Highlights

- Physics-based SHM systems produce a high level of accuracy of structural response prediction; however, they require domain expertise to define an FEM model with correct boundary conditions, constraints and structural properties.
- For long-term monitoring systems and in operation, a strategy for updating the FEM model should be defined in order to have a model that reflects and simulates the

current state of the structure. This strategy should define the update method, which parameters to update and when to update the structural parameters.

- To define the initial FEM model of the structure of interest, a stage of identification of the structural parameters must be considered, using estimation algorithms that include wear effects, operating conditions, the uncertainty of material properties, as well as the possibility of underlying structural damage. Using a data-driven method for the identification of these parameters, through a regression task by a machine-learning algorithm is recommended.
- For real-time SHM applications based on physical models, optimization algorithms in the solution of the numerical computation problem of an FEM model must be implemented to guarantee the responsiveness and availability of the SHM system to an event or in the continuous monitoring task.

Derived from the analysis of the works shown above in Tables 8–10, the most relevant observations of the application of ML techniques are presented below.

### 6.2. ANN SHM Highlights

- The ANN-based SHM systems showed good damage identification results using the modal information extracted from the structural acceleration responses with less training time compared to the proposed CNN and SVM, and simplicity in implementation.
- The selection of damage-sensitive features include the use of natural frequencies, modal shapes, damage index based on modal strain energy, FRF estimation, IMF or a combination of them.
- The removal of environmental effects and noise is also encouraged by signal processing techniques, such as filters (Kalman filter), and transforms, such as WT, EMD and HHT. Indeed, natural frequencies are susceptible to effects by temperature changes [23].
- The need for a large number of sensors is a drawback for ANN-based SHM system implementations. Analytical or data-driven estimation of unmeasured locations in the structure can enrich the input data to improve structural damage identification, as shown in [76]. The use of a sensor network allows the application of data fusion techniques that mitigate redundant signal information and anomalous sensor data.

### 6.3. CNN SHM Highlights

- The optimization techniques for the design of CNN architectures for SHM systems should be considered. Some popular proposals include optimization algorithms such as PSO and HSA. The design of the CNN architecture in SHM applications is the focus of attention since there is no method to select the hyperparameters of the CNN architecture.
- Two main dimension variants are used in the proposed CNN-based SHM systems: 1D-CNN and 2D-CNN. In the proposals with 1D input in SHM CNN, marginal HHT spectra, raw response time series windows, mode curvature differences, first-order mode shapes and Fourier amplitude spectra (FAS) have been used as input vectors. For 2D input matrices in SHM CNN, time–frequency plots, time series data rearranged into square matrices, multi-sensor matrix series data, heat map matrices and discrete response histograms have been used.
- A large amount of training data are needed to train CNN architectures. FE-generated data can be used, but this generated data can be unreliable if the FE model has uncertainties. Data enrichment can be performed to make the data reliable for training purposes, as shown in [92]. Data augmentation techniques and GAN-generated data can also be used and it is also necessary to balance the training datasets.
- CNN shows reliable performance, especially on noisy datasets [94] using raw measurements, compared to SVM [2] and MLP models [99]. In addition, CNN pre-preserves dominant frequency signal features, eliminating high-frequency noise components [99].

### 6.4. SVM SHM Highlights

- To ensure a good level of generality in structural damage identification capabilities, sufficient examples of damage scenarios should be provided in the model training

step. The data generated by the FEM could be useful in scenarios where the damage state of the building is not available. Classification errors may occur when unlearned pattern scenarios are fed to the SVM model.

- It is essential to keep the number of SVM features to a computationally feasible number due to the computational processing cost. To reduce this cost, dimensionality reduction techniques such as PCA, random projection and boosting techniques are encouraged.
- The performance of the final SVM depends critically on the damage-sensitive features selected in the model input. Popular selected damage features include ARM features, wavelet-based energy features, and statistical features. If denoising techniques are not applied, the final performance may also be affected [45].
- There is a high dependence between the selected hyperparameters of the SVM and its performance, so the use of optimization techniques for hyperparameter tuning, such as PSO, PSH, CSA and GA, is recommended. More recent optimization techniques include sunflower optimization [144].
- The model training time using SVM approaches might be inadequate for model update stages in dynamic SHM building models and real-time building monitoring systems.
- SVM approaches show high capability for small structural damage identification [104] in combination with feature selection techniques.

## 7. Conclusions

This study presented a review of recent solutions in the field of SHM systems for multi-story buildings and bridges, addressing areas of improvement, limitations and opportunities. The limitations of physics-based and data-driven models in SHM in buildings and bridges were also discussed. This review showed the popularity of SHM systems based on the ML techniques of SVM, NN, and CNN due to the benefits of these algorithms in classification capabilities. The advantages and disadvantages of applying these ML techniques were also discussed in order to provide valuable insight for researchers.

In general, a fair comparison between ML models and feature extraction techniques implies a similar definition of damage scenarios (i.e., loss of stiffness, bolt loosening or mass addition) to provide a reasonable comparison. The occurrence of multiple damage locations within the structure is also an underexplored area in damage scenarios.

Real-time monitoring with in-service structure systems also needs to be developed. In addition, anomalous sensor data must be taken into account in real-time applications. Real-world structures are more complex than some idealized FE models and must involve a higher degree of uncertainty than those idealized models, so more damage scenarios must be studied to verify damage identification. FE model updating techniques and ML model retraining techniques should be explored for long-term SHM systems in the construction stages of the operational phase. Sensor placement plays a crucial role in the performance of ML-based SHM systems, so more studies should be conducted analyzing this fact.

The implications due to constraints imposed by the operational and environmental effects on the model need to be explored in implementation scenarios. For example, the periodicity, in which data should be collected from the monitored system to adjust or update the model due to the degradation of physical properties or operational conditions. Signal processing techniques, such as data fusion, normalization and compression, could improve data quality. In addition, with the development of increasingly powerful devices with higher computing power, signal processing can be performed in the fog computing paradigm. Fog devices can speed up the process of noise filtering, data compression and data fusion in SHM systems to assess the state of the structure after a seismic event. Hybrid approaches that integrate physics-based and data-based models are encouraged to address some of the drawbacks of stand-alone modeling, and this is the approach taken by some of the most recent developments in the SHM field. The need for an SHM system that synergistically integrates the advances reported in the literature in data generation, sensing, processing, model parameter updating and real-time monitoring, including the rapid assessment of post-seismic event response, is envisioned.

**Author Contributions:** Conceptualization, A.G.-C. and P.J.E.-A.; investigation, A.G.-C.; writing—original draft preparation, A.G.-C.; writing—review and editing, P.J.E.-A.; supervision, P.J.E.-A. All authors have read and agreed to the published version of the manuscript.

**Funding:** This work was funded in part by Consejo Nacional de Ciencia y Tecnología (CONACYT) and by Instituto Politécnico Nacional (IPN) under Grant SIP-20221495.

**Institutional Review Board Statement:** Not applicable.

**Informed Consent Statement:** Not applicable.

**Data Availability Statement:** Not applicable.

**Conflicts of Interest:** The authors declare no conflict of interest.

**Abbreviations**

| Definition | Abbreviation | Definition | Abbreviation |
|---|---|---|---|
| Added random noise | ARN | Linear discriminant analysis | LDA |
| Auto-associative neural network | AANN | Mahalanobis-squared distance | MSD |
| Autoregressive model | ARM | Mean absolute error | MAE |
| Back-propagation | BP | Mean-squared error | MSE |
| Back-propagation neural network | BP-NN | Modal assurance criterion | MAC |
| Camage classification rates | DCR | Modified ensemble empirical mode decomposition | MEEMD |
| Complete ensemble empirical mode decomposition | CEEMD | Naïve bayes | NB |
| Contribution damage localization methods | CDLM | Nonlinear time–history analysis | NTHA |
| Convolutional neural network | CNN | Particle swarm optimization | PSO |
| Coupled simulated annealing | CSA | Piecewise aggregate approximation | PAA |
| Cross-correlation coefficient | CC | Poly-reference least-square complex frequency domain | p-LSCF |
| Damage-sensitive energy feature | DEF | Power spectral densities | PSD |
| Decision tree | DT | Principal component analysis | PCA |
| Deep convolutional neural network | DCNN | Principal component pursuit | PCP |
| Deep neural network | DNN | Probabilistic neural network | PNN |
| Differential evolution algorithm | DEA | Quadratic discriminant analysis | QDA |
| Electromechanical impedance | EMI | Radial basis function neural network | RBFNN |
| Extended Kalman filter | EKF | Random projection | RP |
| Factor analysis | FA | Recursive algorithm autoregressive-moving average with the exogenous inputs | RARMX |
| Fast Fourier transform | FFT | Residual error | RE |
| Feed-forward multilayered perceptron | FFMLP | Scaled conjugate gradient algorithm | SCG |
| Fourier transform | FT | Self-organizing maps | SOM |
| Frequency-domain decomposition | FDD | Signal statistical indicators | SSaI |
| Frequency response functions | FRF | Singular value decomposition | SVD |
| Gaussian radial basis function | GRBF | Statistical features | SF |
| Gaussian white noise | GWN | Stiffness reduction factor | SRF |
| Generative adversarial network | GAN | Stochastic subspace identification | SSI |
| Gradient descent momentum | GDM | Support vector data description | SVDD |
| Harmony memory | HM | Support vector machine | SVM |
| Harmony search algorithm | HSA | Teager–Huang transform | THT |
| High-pass filter | HPF | Trainable look-up tables | LUT |
| Hilbert–Huang transform | HHT | Transmissibility function analysis | TFA |
| Imperial competitive algorithm | ICA | Unsupervised image transformation model | UITM |
| Improved reduction system | IRS | Wavelet package relative energy | WPRE |
| Kernel principal component analysis | KPCA | Wavelet packet decomposition | WPD |
| Least-square support vector machine | LS-SVM | Wavelet packet transform | WPT |
| Levenberg–Marquardt back-propagation | LMBP | Wavelet-based autoregressive model | WARM |

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
