# Peer review of "Review of Machine-Learning Techniques Applied to Structural Health Monitoring Systems for Building and Bridge Structures"

_applsci, doi:10.3390/app122110754_

Round 1

Reviewer 1 Report (New Reviewer)

General comments:

The paper presents a very detailed review about current machine learning algorithms implemented in vibration-based building structural health monitoring systems. The application of three machine learning techniques, namely artificial neural networks (ANN), convolutional neural networks (CNN), and support vector machines (SVM) techniques are compared according to the input data, feature selection techniques, structure of interest, data size, level of damage detection and accuracy of the ML model. The paper addresses recent developments and limitations of SHM systems, and presents some promising future work.

Specific comments:

1.        The author uses the term “damage detection” to denote the whole process of damage characterization, but it is very confusing since damage detection is also the first level of damage characterization as shown in Table 1. Maybe it is better to use “damage diagnosis” or “damage identification” to represent the whole process.

2.        In Ref. [6], Farrar and Worden point out 5 levels of damage identification, namely detection, localization, classification, assessment, and prediction, but why there are just 4 levels included in Table 1?

3.        In section 4.2, the author should add a brief description of unsupervised learning in SHM applications. Unsupervised learning is vitally important in SHM although it could only achieve damage detection and localization tasks since labels of damage scenarios is sometimes difficult to obtain, especially for large complex structures.

4.        As mentioned in the abstract, uncertainty treatment is vitally important in machine learning algorithms to provide a robust result under modelling error and measurement noise. Therefore, it would be better to add a section focusing on this aspect. For example, there are some works adopting probabilistic distance of transmissibility to accommodate the uncertainties involved in structural anomaly detection published in JSV and MSSP, which are suggested to be included for the sake of completeness of survey.

Author Response

Original Manuscript ID:  applsci-1932978
Original Article Title: “ Review of Machine Learning Techniques Applied to Structural Health Monitoring Systems for Building and Bridge Structures. ”

To:  Applied Sciences MDPI journal Special Issue Editor
Re: Response to reviewers

Dear Editor,

Thank you for allowing a resubmission of our manuscript, with an opportunity to address the reviewers’ comments.

We are uploading (a) our point-by-point response to the comments (below) (response to reviewers), (b) an updated manuscript with yellow highlighting indicating changes, and (c) a clean updated manuscript without highlights (PDF main document).

Best regards,
Ponciano Jorge Escamilla-Ambrosio,  Alain Gomez-Cabrera

Reviewer #1, Concern # 1: The author uses the term “damage detection” to denote the whole process of damage characterization, but it is very confusing since damage detection is also the first level of damage characterization as shown in Table 1. Maybe it is better to use “damage diagnosis” or “damage identification” to represent the whole process.

Author response: We acknowledge reviewer’s comment. The term damage detection is confusing when used to describe the general process of damage characterization.

Author action: We have updated the whole manuscript by changing “damage detection” to “damage identification” to denote the whole process of damage characterization.

Reviewer #1, Concern # 2:  In Ref. [6], Farrar and Worden point out 5 levels of damage identification, namely detection, localization, classification, assessment, and prediction, but why there are just 4 levels included in Table 1?

Author response: We acknowledge reviewer’s comment. As the reviewer mentions, the scheme mentioned does not correspond to the reference. The structural damage identification classification scheme chosen in this study corresponds to the proposal presented by Ritter et al. and not to the one presented by Farrar and Worden. This is an error and should be corrected.

Author action: We have updated the manuscript by replacing the reference 6 by the following reference that contains the correct four-level damage identification scheme.

Rytter, A. Vibrational based inspection of civil engineering structures. 1993.

Also we have updated the manuscript by fixing the author name in the reference paragraph:

Rytter [7] also presents a damage identification classification in SHM systems as shows in Table 1.

Reviewer #1, Concern # 3: In section 4.2, the author should add a brief description of unsupervised learning in SHM applications. Unsupervised learning is vitally important in SHM although it could only achieve damage detection and localization tasks since labels of damage scenarios is sometimes difficult to obtain, especially for large complex structures.

Author response: We acknowledge reviewer’s comment. We have decided to add a short section to the paper mentioning some work related to the application of unsupervised machine learning methods in the area of SHM.

Author action: We have updated the manuscript by adding the following section:

4.2.4 Unsupervised Learning Algorithms

Unlike the machine learning methods discussed in the previous sections, unsupervised learning-based methods use training data samples without the need for class labels. Intuitively, these types of methods avoid the problem of counting possible multiple structural damage cases, including multiple structural damage scenarios. By applying unsupervised methods, two types of detections can be performed from the data obtained by the sensor network measurements of SHM systems: novelty detection and anomaly detection.

To address the drawbacks of supervised machine learning approaches applied to structural damage identification, including sample labeling and the need for data from multiple damage scenarios, Wang and Cha [50] presented an unsupervised ML approach that uses an autoencoder to extract features from the raw measurement data. An SVM was trained with the extracted features, allowing to detect even small damages (<10% stiffness reduction) in a structural member of a steel bridge.

Entezami and Shariatmadar [127] propose two new damage indices for damage localization and quantification from time series modeling using an AR model. In this approach, thresholds are defined to detect possible structural damage without establishing damage patterns. Daneshvar and Sarmadi [128] explore an unsupervised anomaly-based method for short- and long-term SHM using an anomaly score and also setting an alarm threshold to establish damage. The core of the method implements an unsupervised feature selection through a one-class nearest neighbor rule.

De Almeida Cardoso et al [129] proposed a damage detection framework based on outlier de-detection in the acquired data. The data descriptors used in this proposal include expected data value, data variability, data symmetry, and data flatness. In this proposal, k-medoids clustering technique to detect outlier data using a proposed symbolic distance voting scheme. Eltouny and Liang [130] propose a post-earthquake damage detection and localization approach that combines intensive cumulative measurements with an unsupervised learning algorithm. This approach consists of two stages: in the first stage, a feature extraction process is performed to extract the cumulative measures and in the second stage, joint probabilities are used as damage indicators obtained through the kernel multivariate maximum entropy method.

This review will emphasize supervised learning methods such as ANN, CNN and SVM in combination with preprocessing techniques such as WT, EMD, HHT and PCA. Despite the advantages of using unsupervised techniques in the problem of structural damage identification, the use of supervised techniques is the most adopted branch in the literature.

Reviewer #1, Concern # 4: As mentioned in the abstract, uncertainty treatment is vitally important in machine learning algorithms to provide a robust result under modelling error and measurement noise. Therefore, it would be better to add a section focusing on this aspect. For example, there are some works adopting probabilistic distance of transmissibility to accommodate the uncertainties involved in structural anomaly detection published in JSV and MSSP, which are suggested to be included for the sake of completeness of survey.

Author response: We acknowledge reviewer’s comment.

Author action: We have updated the manuscript by adding

  1. Uncertainties effects minimization in SHM systems

Within the training and operation process of SHM systems, model uncertainties and monitoring system anomalies adversely affect performance and damage detection capability.  These uncertainties can be found in the data that are used for training and in the data that are processed during system operation.

In the training phase of the machine learning model, it is common to use synthetic training data generated from FEM models of the structure of interest. Simulation of the structural response facilitates the task of generating data under different operating and structural damage conditions. The initial model of the structure of interest must accurately reflect the structural behavior and response, as it is the starting point for comparison with future states to determine the presence of structural damage. To minimize the uncertainties of the structural FEM model, several solutions have been proposed in recent years to obtain a model with reliable parameters. Several solutions have been proposed for the optimal way to perform FEM model updating in the SHM area.

Lee and Cho [131] proposed a direct FEM update strategy based on the identification of modal parameters and their comparison with the predicted structural response from measurements on a bridge structure. Jafarkhani [132] presented an evolutionary strategy in combination with a preliminary damage detection update triggering process and an ARMA model. An iterative FEM model updating process is used in Pachon's approach [133] where a sensitivity analysis determines the most significant structural parameters that have influence on the overall behavior of the structure. A survey of different FEM model updating methods in the area of SHM can be found in [134]. In the case of bridge-type structures, an analysis of the different model updating techniques can be found in Sharry's work in [135]

Other approaches in the area of unsupervised learning that consider the effect of uncertainties in measured data have been proposed for anomaly detection systems that are resistant to uncertainty in sensor data. Yan et al. [141] studied the use of symmetric Kullback-Leibler (SKL) distance and transmissibility function to perform structural anomaly detection by comparing a baseline state and potential damage scenarios. Bayesian inference and Mon-te Carlo discordance test are used as a statistical screening scheme to deal with measurement uncertainties. To improve the probabilistic function of the anomaly detection model, Mei et al. [142] use the Bhattacharyya distance of TF. The performance of this proposal outperforms other proposals using Mahalanobis distance based methods and have better resistance against the effects of uncertainties. Sarmadi and Karamodin [143] present a Mahalanobis-squared distance-based method that combines the one-class kNN rule and an adaptive distance measure. In this study, the clustering process eliminates the effects of varying environmental conditions on the anomaly detection process.

Reviewer 2 Report (New Reviewer)

General comments

In a general way, the review paper identifies current machine learning algorithms implemented in building structural health monitoring systems and their success in determining the level of damage in a hierarchical classification. The text is well structured and present relevant content on the SHM field. However, there are some parts that must be revised for better understanding.

Specific comments and technical remarks:

1.    The authors must improve and highlight some quantitative results at the end of the abstract.

2.    Section 1: last paragraph: what is the research gap? What do you propose with this “new” review paper?

3.    Section 2 to 5: authors could improve the readability by including some figures (examples, etc.) to represent the method.

4.    The authors could improve the references with new (2022) studies: 10.1007/s12065-021-00652-4, 10.1007/s00170-018-2502-z, 10.1016/j.engappai.2020.104055.

The authors highlighted many numerical methods in the manuscript. There is no conclusion about this. Is there any preference or advantage in any of the highlighted methods? Authors must discuss this and highlight the advantages and disadvantages.

Author Response

Original Manuscript ID:  applsci-1932978
Original Article Title: “ Review of Machine Learning Techniques Applied to Structural Health Monitoring Systems for Building and Bridge Structures. ”

To:  Applied Sciences MDPI journal Special Issue Editor
Re: Response to reviewers

Dear Editor,

Thank you for allowing a resubmission of our manuscript, with an opportunity to address the reviewers’ comments.

We are uploading (a) our point-by-point response to the comments (below) (response to reviewers), (b) an updated manuscript with yellow highlighting indicating changes, and (c) a clean updated manuscript without highlights (PDF main document).

Best regards,
Ponciano Jorge Escamilla-Ambrosio,  Alain Gomez-Cabrera

Reviewer #2, Concern # 1: The authors must improve and highlight some quantitative results at the end of the abstract.

Author response: We acknowledge reviewer’s comment. we have decided to add the number of articles corresponding to the supervised machine learning methods as well as the time period considered as a cut-off for the literature review in the abstract section.

Author action: We have updated the manuscript by adding the following lines in the abstract:

A total of 68 articles using ANN, CNN and SVM in combination with preprocessing techniques were analyzed corresponding to the period 2011-2022.

Reviewer #2, Concern # 2:  Section 1: last paragraph: what is the research gap? What do you propose with this “new” review paper?

Author response: We acknowledge reviewer’s concern. Although there are previous studies analyzing different proposals in the area of SHM, the study we present here is limited to civil structures such as buildings and bridges. Additionally, this review collects and analyzes only those systems that make use of acceleration signals as data used for damage identification tasks. Within the document, the following paragraph in the Introduction Section highlights the characteristics of this review as well as the added value pursued:                                                                                                            

Previous studies have been conducted in the field of SHM using machine learning (ML) techniques. However, the importance of this study is to compare the application of artificial neural networks (ANN), convolutional neural networks (CNN) and support vector machines (SVM) techniques considering the input data, feature selection techniques, structure of interest, data size, level of damage identification and accuracy of the ML model. In addition to the fact that each of the above ML techniques has been used for similar tasks in structural damage and system identification, some of them perform better when the data comes from data generated by multi-sensor data fusion or when the data is processed with damage-sensitive feature extraction techniques such as Hilbert–Huang transform (HHT) or wavelet packet transform (WPT). In addition, this study could provide a starting point for the selection of ML techniques and signal processing techniques for future SHM ML-based solutions where similar structural configuration or data features have similarity with previous studies with good structural damage or system identification performance

Author action: No action taken.

Reviewer #2, Concern # 3: Section 2 to 5: authors could improve the readability by including some figures (examples, etc.) to represent the method.

Author response: We acknowledge reviewer’s comment. We recognize that the article has very few illustrative images (in fact, only one figure in the original manuscript) so we have decided to add additional figures.

Author action: We have enriched the manuscript by adding the following 3 figures:

Figure 2. Multi-layer hidden layer feedforward ANN

Figure 3. A CNN network example using raw acceleration input data.

Figure 3. A SVM-based structural damage identification system example [114]

Reviewer #2, Concern # 4:  The authors could improve the references with new (2022) studies:

10.1007/s12065-021-00652-4,

10.1007/s00170-018-2502-z,

10.1016/j.engappai.2020.104055.

Author response: We acknowledge reviewer’s comment. The articles mentioned by the reviewer enrich the manuscript and we have decided to add them to exemplify optimization using bio-inspired algorithms.

Author action: We have updated the manuscript by adding the following in-text references:

Gomes et al. [56] addresses an inverse identification problem using numerical models and a genetic algorithm.

In order to reduce the time costs of performing calculations, optimization algorithms applied to the problem of SHM are encouraged and proposed in the literature [57].

More recent optimization techniques include sunflower optimization [144]

Also we have updated the manuscript by adding the following references:

Gomes, G.F.; de Almeida, F.A.; da Cunha, S.S.; Ancelotti, A.C. An estimate of the location of multiple delaminations on aeronautical CFRP plates using modal data inverse problem. The International Journal of Advanced Manufacturing Technology 2018, 99, 1155-1174, 1051 doi:10.1007/s00170-018-2502-z.

Pereira, J.L.J.; Francisco, M.B.; Cunha Jr, S.S.d.; Gomes, G.F. A powerful Lichtenberg Optimization Algorithm: A damage identification case study. Engineering Applications of Artificial Intelligence 2021, 97, 104055, doi:https://doi.org/10.1016/j.engappai.2020.104055.

Magacho, E.G.; Jorge, A.B.; Gomes, G.F. Inverse problem based multiobjective sunflower optimization for structural health 1233 monitoring of three-dimensional trusses. Evolutionary Intelligence 2021, doi:10.1007/s12065-021-00652-4.

Reviewer #2, Concern # 5:  The authors highlighted many numerical methods in the manuscript. There is no conclusion about this. Is there any preference or advantage in any of the highlighted methods? Authors must discuss this and highlight the advantages and disadvantages.

Author response: We acknowledge reviewer’s comment. To complement the Highlights section of the document we have added a list of key points concerning the application of physical models using numerical methods such as FEM models.

Author action: We have updated the manuscript by adding the following paragraphs

6.1 Physics-based SHM highlights

  • Physics-based SHM systems produce a high level of accuracy of structural response prediction, however they require domain expertise to define a FEM model with correct boundary conditions, constraints and structural properties.
  • For long-term monitoring systems and in operation, a strategy for updating the FEM model should be defined in order to have a model that reflects and simulates the current state of the structure. This strategy should define the update method, which parameters to update and when to update the structural parameters.
  • To define the initial FEM model of the structure of interest, a stage of identification of the structural parameters must be considered, using estimation algorithms that include wear effects, operating conditions, uncertainty of material properties as well as the possibility of underlying structural damage. Using a data-driven method for the identification of these parameters through a regression task by a machine learning algorithm is a solution for this task.
  • For real-time SHM applications based on physical models, optimization algorithms in the solution of the numerical computation problem of a FEM model must be implemented to guarantee the responsiveness and availability of the SHM system to an event or in the continuous monitoring task.

Round 2

Reviewer 2 Report (New Reviewer)

The authors performed all the recommended corrections.

This manuscript is a resubmission of an earlier submission. The following is a list of the peer review reports and author responses from that submission.

Round 1

Reviewer 1 Report

See attached document below

Reviewer 2 Report

The paper "Review of Machine Learning Techniques Applied to Structural Health Monitoring Systems for Building Structures" it presented as a review article and, as such, significant novelty or scientific soundness are not expected.

The paper is, however, well written and presents in a smart way a possible classification of machine learning techniques applied to SHM.  

However, the authors could make some effort to improve paragraphs 6 and 7 in which very sensitive aspects such as damage factors to be considered and environmental effects on measurements are discussed. Improvement is expected in terms of providing a more "critical review" from which, also through some revision in the conclusion paragraph, the reader could witness some expert contribution by the authors in not only aseptically reporting what other scientists have published on the topic.  

Reviewer 3 Report

The title is a review of machine learning (ML) methods in building structural health monitoring (SHM), but the organizational structure and content do not match the title. Specifically, the review about ML in building SHM in the paper is not ample. In addition, there is no need for a large repetition of the basic common sense regarding damage classification in SHM. There should be more discussions on the ML methods reported in building SHM. As far as the title of the paper is concerned, ‘bridge’ in subtitle 6 is confusing. Moreover, given the rapid development of deep learning technology in the field of machine learning, focusing on deep learning methods used in building SHM is more meaningful.